# Adipose tissue-derived neurotrophic factor 3 regulates sympathetic innervation and thermogenesis in adipose tissue

Xin Cui [1], Jia Jing[1], Rui Wu[1], Qiang Cao[1], Fenfen Li[1], Ke Li [1], Shirong Wang[1], Liqing Yu[2], Gary Schwartz [3], Huidong Shi [4,5], Bingzhong Xue [1✉] & Hang Shi[1✉]

Activation of brown fat thermogenesis increases energy expenditure and alleviates obesity. Sympathetic nervous system (SNS) is important in brown/beige adipocyte thermogenesis. Here we discover a fat-derived "adipokine" neurotrophic factor neurotrophin 3 (NT-3) and its receptor Tropomyosin receptor kinase C (TRKC) as key regulators of SNS growth and innervation in adipose tissue. NT-3 is highly expressed in brown/beige adipocytes, and potently stimulates sympathetic neuron neurite growth. NT-3/TRKC regulates a plethora of pathways in neuronal axonal growth and elongation. Adipose tissue sympathetic innervation is significantly increased in mice with adipocyte-specific NT-3 overexpression, but profoundly reduced in mice with TRKC haploinsufficiency (TRKC +/−). Increasing NT-3 via pharmacological or genetic approach promotes beige adipocyte development, enhances cold-induced thermogenesis and protects against diet-induced obesity (DIO); whereas TRKC +/− or SNS TRKC deficient mice are cold intolerant and prone to DIO. Thus, NT-3 is a fat-derived neurotrophic factor that regulates SNS innervation, energy metabolism and obesity.

[1] Department of Biology, Georgia State University, Atlanta, GA 30303, USA. [2] Department of Medicine, University of Maryland School of Medicine, Baltimore, MD 21201, USA. [3] Department of Medicine, Albert Einstein College of Medicine, Bronx, NY 10461, USA. [4] Georgia Cancer Center, Augusta University, Augusta, GA 30912, USA. [5] Department of Biochemistry and Molecular Biology, Medical College of Georgia, Augusta University, Augusta, GA 30912, USA. ✉email: bxue@gsu.edu; hshi3@gsu.edu

Obesity has become a serious health problem that poses as a major risk factor for the development of a panel of metabolic diseases such as insulin resistance/type 2 diabetes, dyslipidemia, hypertension, and cardiovascular diseases[1]. Obesity is caused by a chronic energy imbalance that results from energy intake over energy expenditure[1]. Adipose tissue is one of the most important organs that regulate energy homeostasis in the body[2]. There exist two distinct types of adipose tissues: white adipose tissue (WAT) that is specialized in energy storage and brown adipose tissue (BAT) that is unique for energy dissipation[2,3]. BAT has the capacity to dissipate energy through adaptive thermogenesis, a biological process in which energy is burned as heat instead of being trapped in ATP[2,4]. The thermogenic activity of BAT largely depends on the unique action of uncoupling protein 1 (UCP1) in the inner mitochondrial membrane, which actively uncouples oxidative phosphorylation from ATP synthesis, thereby profoundly increasing energy expenditure[2,5]. However, recent studies also discovered UCP1-independent thermogenesis mediated by SERCA2b-induced calcium cycling and creatine-driven substrate cycling[6,7]. Brown fat thermogenesis was traditionally viewed as a defense mechanism against cold in rodents. A recent discovery of metabolically active brown fat in adult humans has further implicated BAT thermogenesis as a promising therapeutic target for the treatment of obesity[8–10].

There are two kinds of UCP1-positive brown adipocytes identified in rodents: traditional brown adipocytes residing in an anatomically defined area (e.g., interscapular (iBAT)) and beige adipocytes dispersed in white fat depots[2,5]. Unlike brown adipocytes that develop prenatally and persist throughout lifetime, beige adipocytes are mostly induced by cold and β-adrenergic agonists[2,5]. Xue et al. previously reported the existence of a unique subset of beige adipocytes, so-called developmentally induced beige adipocytes, which are transiently induced in postnatal mice[11].

Adipose tissue is innervated by SNS that plays a key role in adipose lipolysis and BAT/beige thermogenesis[12–16]. Catecholamines released by sympathetic nerve terminals in response to cold stimulate lipolysis and activate BAT/beige thermogenesis via β-adrenergic receptors[17,18]. Although extensive studies have been devoted to the role of neuronal network and diverse neural-endocrine pathways in the regulation of SNS activation[19,20], much less is known about the role of SNS-targeted tissues (e.g., BAT and WAT) in regulating the development and activation of SNS. Recent data demonstrated that S100, a BAT-derived secretory protein, promotes SNS innervation into adipose tissue[21]. However, the mechanism underlying the neurotrophic effect of S100 on SNS is not entirely clear.

It is noteworthy that the developmentally induced beige adipocytes appear transiently, peaking at postnatal day 20 and then disappearing thereafter toward adulthood[11]. However, the mechanisms mediating the induction and disappearance of the developmental beige adipocytes are not known. Here, we found that the disappearance of the developmental beige cells in adult mice is associated with the diminished sympathetic innervation in white adipose tissue. Our data indicate that the physiological pathway(s) required to maintain the vigorous innervation of sympathetic nerves could decline in adult mice, leading to the disappearance of the developmental beige adipocytes. We discovered that the expression of the fat-derived neurotrophic factor NT-3 is higher in mouse BAT than WAT; NT-3 expression coincides with the appearance of developmentally induced beige adipocytes in postnatal mice and is induced in WAT by cold exposure in adult mice. Using pharmacological and genetic approaches, we further determined the role of NT-3 and its receptor neurotrophic receptor 3/Tropomyosin receptor kinase C (TRKC) in the regulation of SNS growth and innervation in

adipose tissue and energy metabolism during the cold- and diet-induced thermogenesis.

## Results

**Adipose-derived neurotrophic factor NT-3 correlates with UCP1 expression and sympathetic innervation in adipose tissue.** Xue et al. (a senior author in this paper) previously found that beige adipocytes can be induced in WAT not only during cold exposure in adult animals but also in newborn pups, which peaked at 20 days of age[11]. These beige adipocytes then gradually disappeared and replaced by mature white adipocytes by the time when mice were 2 months of age[11]. Indeed, here we found that Ucp1 mRNA in inguinal WAT (iWAT) was profoundly reduced in 3-month- and 6-month-old mice compared to that of postnatal day 20 (P20) pups (Fig. 1a). A similar reduction of UCP1 protein levels in iWAT was observed in 3-month-old adult mice compared to P20 pups (Fig. 1b). The reduction of UCP1 expression in adult iWAT was associated with reduced expression of the SNS marker tyrosine hydroxylase (TH) measured by immunoblotting (Fig. 1b). A similar reduction of UCP1 and TH protein levels was also observed in epididymal WAT (eWAT) of 3-month-old adult mice compared to that of P20 pups (Supplementary Fig. 1a). We also established a whole-mount adipose tissue clearing approach (Adipo-Clear) to permit more comprehensive and accurate three-dimensional visualization of sympathetic innervation with immunostaining of TH[22]. Compared to iWAT from P20 pups, iWAT from 3-month-old mice had significantly diminished SNS nerve innervation (Fig. 1c, left panel). Quantitation of SNS innervation in iWAT using Imaris Image Analysis Software revealed that mean nerve fiber density, mean nerve fiber length, and mean nerve branching points were significantly reduced in iWAT of 3-month-old mice compared to that of P20 pups (Fig. 1c, right panel). A similarly diminished sympathetic innervation was also observed in eWAT of 3-month-old adult mice compared to that of P20 pups (Supplementary Fig. 1b). These data suggest that the reduced SNS innervation may lead to the disappearance of the developmental beige adipocytes in adult mice.

On the other hand, UCP1 and TH protein in iBAT was only slightly decreased in 3-month-old adult mice compared to that of P20 pups (Supplementary Fig. 1c), suggesting that unlike WAT, UCP1 expression and sympathetic innervation in adult iBAT is largely preserved.

SNS innervation in adipose tissue is known to regulate adipocyte lipolysis[14,23]. Interestingly, whereas there was no difference in serum non-esterified fatty acids (NEFA) level between 20-day- and 3-month-old mice housed at room temperature, serum NEFA level was significantly reduced in 3-month-old mice compared to that of 20-day-old mice after a 6-h cold exposure (Supplementary Fig. 1d). This corresponded to a slightly reduced hormone-sensitive lipase (HSL) phosphorylation (pHSL) in iBAT and a more profoundly reduced pHSL in iWAT of 3-month-old mice after the acute cold exposure as compared to that of 20-day-old mice (Supplementary Fig. 1e). Thus, our data suggest that reduced sympathetic innervation in the fat tissues in older mice is functionally important, as it reduces adipose tissue lipolytic capacity in response to a cold challenge.

Existing literature supports an important role of neurotrophic factor NT-3 in the regulation of normal SNS neuron function and growth and target tissue innervation[24,25]. We surveyed the expression patterns of neurotrophic factors across mouse tissues. We found Nt-3 was ubiquitously expressed in many major tissues, among which iBAT was one of the tissues with the highest Nt-3 expression; whereas Nt-3 expression in iWAT was much lower (Fig. 1d). In contrast, the expression of other members of the

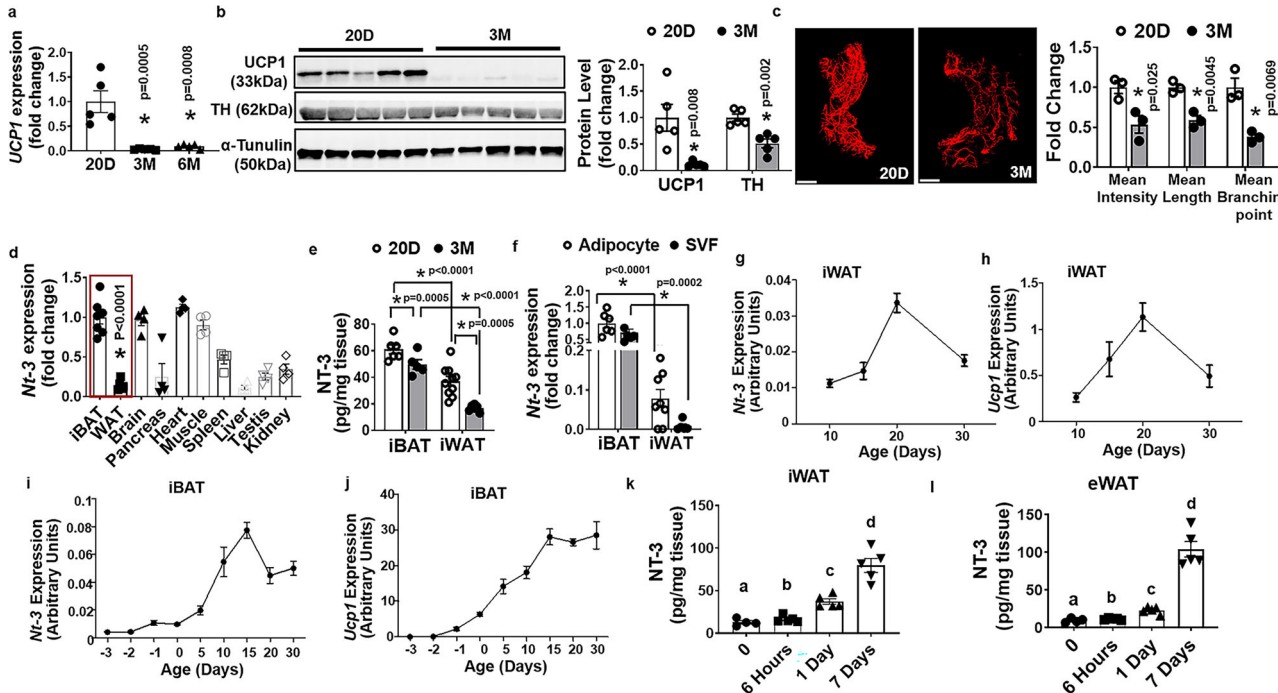

**Fig. 1 Adipose-derived neurotrophic factor NT-3 correlates with UCP1 expression and sympathetic innervation in adipose tissue. a** *Ucp1* mRNA expression in iWAT of 20-day-old postnatal pups, and 3- and 6-month-old mice ($n = 5$/group, one-way ANOVA, $F (2,12) = 18.04$, $P = 0.0002$, *indicates statistical significance vs. 20D with Turkey's multiple comparisons test). **b** UCP1 and TH protein levels in iWAT of 20-day-old postnatal pups and 3-month-old adult mice ($n = 5$/group, * indicates statistical significance vs. 20D using unpaired two-tailed t test). **c** Representative images of TH-positive sympathetic nerve innervation in iWAT (left panel, from three replicates/group, scale bar = 2000 μm) and quantitation of mean nerve fiber density, mean nerve fiber length, and mean branching points normalized to total adipose tissue area (right panel) in iWAT of 20-days-old postnatal pups and 3-month-old adult mice ($n = 3$/group, * indicates statistical significance vs. 20D using unpaired two-tailed t test). **d** Tissue distribution of *Nt-3* expression (iBAT $n = 7$; iWAT $n = 5$; others $n = 4$) (*$P < 0.0001$ iWAT vs iBAT with unpaired two-tailed t test). **e** NT-3 protein levels measured by ELISA in iBAT and iWAT of 20-day-old postnatal pups and 3-month-old adult mice (iBAT 20D $n = 6$, 3 M $n = 5$; iWAT 20D $n = 10$, 3 M $n = 6$) (two-way ANOVA, 20D vs 3 M, $F (1, 24) = 62.2$, $P < 0.0001$; iBAT vs iWAT, $F (1, 24) = 21.95$, $P < 0.0001$. * indicates statistical significance as shown in (**e**) with Turkey's multiple comparisons test). **f** The expression of *Nt-3* mRNA in adipocytes and stromal vascular fraction (SVF) in iBAT and iWAT (iBAT $n = 6$; iWAT adipocyte $n = 8$, SVF $n = 5$, two-way ANOVA, cell type (adipocyte vs SVF), $F (1, 21) = 4.118$, $P = 0.055$; tissue (iBAT vs iWAT), $F (1, 21) = 85.65$, $P < 0.0001$. * indicates statistical significance as shown in (**f**) with Turkey's multiple comparisons test). **g, h** The expression of *Nt-3* (**g**) and *Ucp1* (**h**) in iWAT during postnatal development in mice ($n = 4$/group). **i, j** The expression of *Nt-3* (**i**) and *Ucp1* (**j**) in iBAT during postnatal development in mice ($n = 4$/group). **k, l** NT-3 protein levels measured by ELISA in iWAT (**k**, time 0 $n = 4$; 6 h, 1d and 7d $n = 5$) and eWAT (**l**, time 0 $n = 4$; 6 h, 1d and 7d $n = 5$) during cold exposure. **k** One-way ANOVA with Turkey's multiple comparisons test, $F (3, 15) = 39.79$, $P < 0.0001$. a vs c: time 0 vs 1 day, $P = 0.018$; a vs d: time 0 vs 7 days, $P < 0.0001$; b vs c: 6 h vs 1 day, $P = 0.035$; b vs d: 6 h vs 7 day, $P < 0.0001$; c vs d: 1 day vs 7 days, $P < 0.0001$. **l** One-way ANOVA with Turkey's multiple comparisons test, $F (3, 15) = 64.81$, $P < 0.0001$. a vs c: time 0 vs 1 day, $P = 0.0047$; a vs d: time 0 vs 7 days, $P = 0.0037$; b vs c: 6 h vs 1 day, $P = 0.012$; b vs d: 6 h vs 7 days, $P = 0.0039$; c vs d: 1 day vs 7 days, $P = 0.0069$. All data are expressed as mean ± SEM.

neurotrophin family, nerve growth factor (*Ngf*), brain-derived neurotrophic factor (*Bdnf*), and neurotrophin-4/5 (*Nt4/5*) was generally low and largely comparable in iBAT and iWAT (Supplementary Fig. 2). Consistent with the RNA expression, the NT-3 protein level was also much higher in iBAT than in iWAT, and exhibited a significant decrease in both iBAT and iWAT of 3-month-old mice compared to that of P20 pups; the decrease was more profound in iWAT (Fig. 1e). In addition, when measured in isolated adipocytes and stromal vascular fraction (SVF) in iBAT and iWAT, *Nt-3* was primarily expressed in mature adipocytes in iWAT, whereas *Nt-3* expression was comparable between adipocytes and SVF cells in iBAT (Fig. 1f). Nonetheless, *Nt-3* expression was much higher in both adipocytes and SVF in iBAT than those in iWAT (Fig. 1f).

Interestingly, the expression pattern of *Nt-3* in iWAT during postnatal development (Fig. 1g) mimicked that of *Ucp1* (Fig. 1h), peaked at P20, and then gradually decreased. The expression of *Nt-3* in iBAT peaked at 15 days of age and remained at a relatively high level (Fig. 1i), similar to that of *Ucp1* expression pattern in iBAT (Fig. 1j). In addition, cold exposure in adult mice

significantly induced NT-3 protein levels in both iWAT and eWAT in a time-dependent manner (Fig. 1k, l). Thus, our data suggest that NT-3 is important in maintaining BAT and WAT SNS innervation and *Ucp1* expression, and diminished NT-3 signaling in WAT may be responsible for the disappearance of beige adipocytes in adult mice via downregulating SNS activity and innervation in WAT.

**Administration of NT-3 protein promotes beiging in postnatal mice and protects adult mice from diet-induced obesity.** We reasoned that if fat tissue NT-3 level was maintained during early postnatal development, adequate SNS innervation in WAT might be maintained, which could also maintain adequate beige adipocyte phenotype. Indeed, intraperitoneal (i.p.) injection of NT-3 (50 μg/kg body weight) daily in P20 mice for 10 days from P20 to P30 significantly increased *Ucp1* and other BAT-specific gene expression (Fig. 2a), UCP1 and TH protein levels (Fig. 2b) and UCP1-positive beige adipocytes in iWAT (Fig. 2c); whereas NT-3 administration had minimal effects on *Ucp1* expression or UCP1 immunostaining in iBAT (Supplementary Fig. 3a, b), which

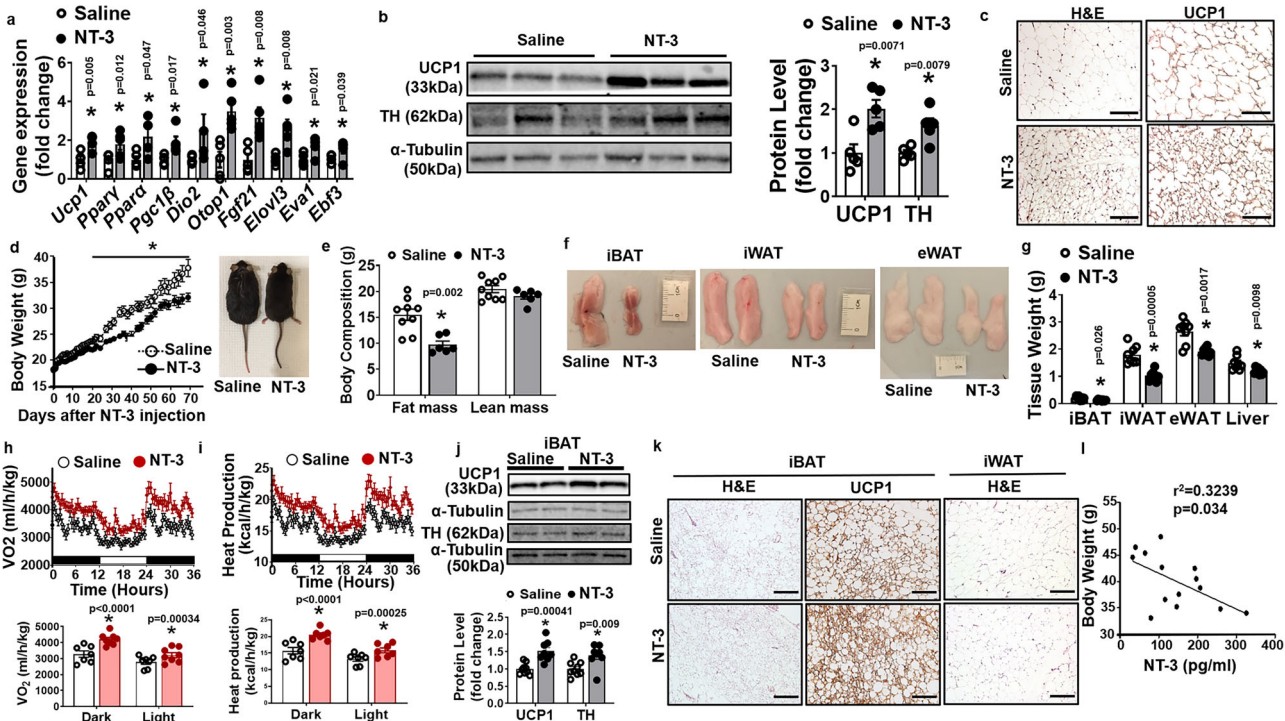

**Fig. 2 Administration of NT-3 protein promotes beiging in postnatal mice and protects adult mice from diet-induced obesity. a–c** *Ucp1* and other BAT-specific gene expression (**a**, n = 5/group, * indicates statistical significance between saline and NT-3 treatments with unpaired two-tailed t test), UCP1 and TH protein levels (**b**, n = 5/group, * indicates statistical significance between saline and NT-3 treatments with unpaired two-tailed t-test) and representative UCP1-immunostaining images showing UCP1-positive beige adipocytes in iWAT (**c**, from three replicate animals/group, scale bar = 150 μm) of 20-day-old postnatal pups with daily intraperitoneal (i.p.) NT-3 (50 μg/kg) injection for 10 days. **d–g** Body weight (**d**, saline n = 9; NT-3 n = 8. * Indicates statistical significance with unpaired two-tailed t test, saline vs NT-3 treatments: D19, P = 0.04; D20, P = 0.038; D21, P = 0.022; D22, P = 0.04; D24, P = 0.03; D27, P = 0.019; D29, P = 0.013; D31, P = 0.013; D34, P = 0.006; D36, P = 0.009; D38, P = 0.009; D40, P = 0.008; D43, P = 0.006; D44, P = 0.006; D45, P = 0.004; D48, P = 0.019; D50, P = 0.027; D52, P = 0.0496; D55, P = 0.026; D57, P = 0.025; D59, P = 0.024; D62, P = 0.018; D64, P = 0.02; D66, P = 0.014; D69, P = 0.008), body composition (**e**, saline n = 9; NT-3 n = 6, * indicates statistical significance between saline and NT-3 treatments with unpaired two-tailed t test), fat pad morphology (**f**) and fat pad mass (**g**, n = 8/group, * indicates statistical significance between saline and NT-3 treatments with unpaired two-tailed t test) in 6-week-old mice with NT-3 (50 μg/kg) i.p. injection every day for the first 2 weeks, then three times a week for 8 weeks while fed HFD. **h, i** Oxygen consumption (**h**) and heat production (**i**) in 6-week-old mice with NT-3 (50 μg/kg) i.p. injection every day for the first 2 weeks, then three times a week for 8 weeks while fed HFD (n = 7/group, * indicates statistical significance between saline and NT-3 treatments with unpaired two-tailed t test). **j, k** UCP1 and TH protein levels in iBAT (**j**, saline n = 9, NT-3 n = 8. * Indicates statistical significance between saline and NT-3 treatments with unpaired two-tailed t test) and representative H&E and UCP1 immunostaining in iBAT and iWAT (**k**, from three replicate animals/group, scale bar for iBAT = 75 μm, and for iWAT=150 μm) of 6-week-old mice with NT-3 (50 μg/kg) i.p. injection every day for the first 2 weeks, then three times a week for 8 weeks while fed HFD. **l** Correlation between serum NT-3 levels and body weight in mice fed a HFD (n = 14, P = 0.034 with simple linear regression and Pearson's correlation test, F (1, 12) = 5.748). All data are expressed as mean ± SEM.

might be expected due to high endogenous NT-3 level in iBAT. These data suggest that continuous activation of NT-3 signaling in iWAT may be important in maintaining beige cell phenotype, and preventing them from disappearing during postnatal development.

To further study the physiological significance of NT-3 in regulating both cold- and diet-induced thermogenesis and energy homeostasis in adult animals, we injected NT-3 (50 μg/kg body weight) into 6-week-old C57BL/6J mice and challenged them with an acute cold exposure (5 °C) or a high-fat diet (HFD). As expected, serum NT-3 level was significantly increased in mice with NT-3 injection (Supplementary Fig. 4a). We found that mice with NT-3 injection were more cold-tolerant during an acute cold exposure, as they can maintain their body temperature better than saline-injected mice (Supplementary Fig. 4b), indicating NT-3-injected mice had enhanced cold-induced thermogenesis. In addition, mice injected with NT-3 were protected from diet-induced obesity as they gained less weight on HFD starting around 5 weeks after NT-3 treatment (Fig. 2d). The NT-3-injected mice showed a reduced fat mass with no difference in

lean mass (Fig. 2e). In consistence, we observed decreased fat pad mass in iWAT, eWAT, and iBAT depots, and decreased liver weight in NT-3-injected mice (Fig. 2f, g). Mice with NT-3 injection had higher oxygen consumption and heat production compared to that of saline-injected mice (Fig. 2h–j), whereas there was no change in the respiratory exchange rate (RER), locomotor activity, and food intake (Supplementary Fig. 4c–e). These data suggest that the reduced adiposity in NT-3-injected mice may be primarily caused by increased energy expenditure. In consistence, we discovered enhanced UCP1 and TH protein expression in iBAT of NT-3-injected mice measured by immunoblotting (Fig. 2j), more UCP1 immunostaining in iBAT (Fig. 2k), and smaller adipocytes in both iBAT and iWAT (Fig. 2k). Mice with NT-3 injection also exhibited lower fed glucose levels (Supplementary Fig. 4f) and improved insulin sensitivity in insulin tolerance (ITT) test (Supplementary Fig. 4g). In addition, there was an inverse correlation between serum NT-3 levels and body weight in mice fed with HFD (Fig. 2l). Thus, our data strongly suggest that NT-3 injection protects mice from diet-induced obesity, primarily due to increased energy expenditure.

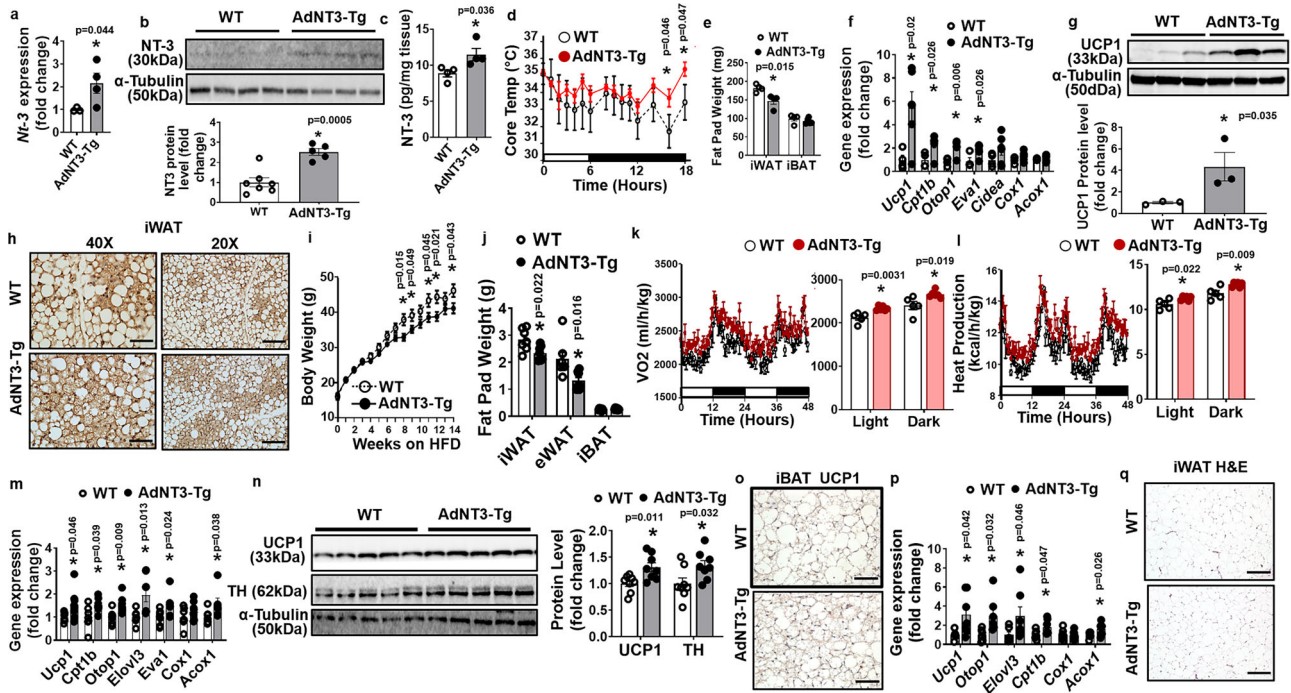

**Fig. 3 Overexpression of NT-3 in adipocytes promotes cold-induced thermogenesis and protects mice from diet-induced obesity. a–c** *Nt-3* mRNA (**a**, WT *n* = 3, AdNT3-Tg *n* = 4, * indicates statistical significance between WT and AdNT3-Tg with unpaired two-tailed *t* test), NT-3 protein levels measured by immunoblotting (**b**, WT *n* = 7, AdNT3-Tg *n* = 5, * indicates statistical significance between WT and AdNT3-Tg with unpaired two-tailed *t* test) and ELISA (**c**, *n* = 4/group, * indicates statistical significance between WT and AdNT3-Tg with unpaired two-tailed *t* test) in iWAT of WT and AdNT3-Tg mice. **d–h** Core body temperature (**d**, *n* = 5/group, * indicates statistical significance between WT and AdNT3-Tg with unpaired two-tailed *t* test), fat pad mass (**e**, *n* = 4/group, * indicates statistical significance between WT and AdNT3-Tg with unpaired two-tailed *t* test), *Ucp1* and other thermogenic gene expression in iWAT (**f**, WT *n* = 4, AdNTF3-Tg *n* = 6, * indicates statistical significance between WT and AdNT3-Tg with unpaired two-tailed *t* test), UCP1 protein levels in iWAT (**g**, *n* = 3, *indicates statistical significance between WT and AdNT3-Tg with unpaired two-tailed *t* test), and representative UCP1 immunostaining images showing UCP1-positive beige adipocytes in iWAT (**h**, from three replicate animals/group, Scale bar for the ×40 images = 75 μm, and for the ×20 images = 150 μm) of AdNT3-Tg mice and their littermates during cold exposure. **i–l** Body weight (**i**, WT *n* = 8, AdNT3-Tg *n* = 10, * indicates statistical significance between WT and AdNT3-Tg with unpaired two-tailed *t* test), fat pad mass (**j**, *n* = 7/group, * indicates statistical significance between WT and AdNT3-Tg with unpaired two-tailed *t* test), Oxygen consumption (**k**, WT *n* = 5, AdNT3-Tg *n* = 6, * indicates statistical significance between WT and AdNT3-Tg with unpaired two-tailed *t* test) and heat production (**l**, WT *n* = 5, AdNT3-Tg *n* = 6, * indicates statistical significance between WT and AdNT3-Tg with unpaired two-tailed *t* test) in AdNT3-Tg mice and their littermate controls fed with HFD starting 6 weeks of age. **m–o** *Ucp1* and other thermogenic gene expression (**m**, WT *n* = 7, AdNTF3-Tg *n* = 8, * indicates statistical significance between WT and AdNT3-Tg with unpaired two-tailed *t* test), UCP1 and TH protein levels (**n**, *n* = 8/group, * indicates statistical significance between WT and AdNT3-Tg with unpaired two-tailed *t* test), and representative UCP1 immunostaining in iBAT (**o**, from three replicate animals/group, scale bar = 75 μm) of AdNT3-Tg mice and their littermate controls fed HFD. **p, q** *Ucp1* and other thermogenic gene expression (**p**, WT *n* = 6, AdNTF3-Tg *n* = 7, * indicates statistical significance between WT and AdNT3-Tg with unpaired two-tailed *t* test), and H&E staining (**q**, from three replicate animals/group, scale bar = 150 μm) in iWAT of AdNT3-Tg mice and their littermate controls fed a HFD. All data are expressed as mean ± SEM.

**Overexpression of NT-3 in adipocytes promotes cold-induced thermogenesis and protects mice from diet-induced obesity.** We have generated a mouse model with NT-3 overexpression specifically in adipocytes under the control of the adiponectin promoter (AdNT3-Tg) (Supplementary Fig. 5a, b), which resulted in an approximately twofold increase in *Nt-3* mRNA levels in iWAT (Fig. 3a), and increased NT-3 protein levels in iWAT as measured by both immunoblotting (Fig. 3b) and ELISA (Fig. 3c). AdNT3-Tg mice also exhibited increased NT-3 protein levels in eWAT and to a lesser extent in iBAT, without changes of NT-3 protein levels in circulation (Supplementary Fig. 5c–e). In contrast, *Nt-3* expression was not altered in other tissues, such as liver and muscle, and in the brain, including arcuate (ARC), paraventricular (PVN), ventromedial (VMH), and lateral (LH) hypothalamus, cortex, and hippocampus (Supplementary Fig. 5f), indicating that *Nt-3* overexpression in AdNT3-Tg mice is specific to adipose tissue. We did not observe much increase in *Nt-3* expression in iBAT of AdNT3-Tg mice, possibly due to high endogenous NT-3 levels in iBAT. In addition, the unchanged

serum NT-3 levels also suggest that NT-3 overexpression in the AdNT3-Tg mice is primarily restricted to adipose tissue due to its mild overexpression in adipose tissue. We, therefore, established an ideal transgenic mouse for the study of the direct effect of the fat-derived NT-3 in sympathetic innervation, which is unlikely confounded by the systemic effect of NT-3 due to the restriction of *NT-3* overexpression to the local adipose tissue.

We first assessed cold-induced thermogenesis in AdNT3-Tg mice. Two-month-old AdNT3-Tg mice and their wild-type (WT) littermate controls were subjected to a cold challenge (5 °C) for 7 days. Interestingly, AdNT3-Tg mice exhibited a tendency of higher body temperature during cold exposure compared to WT, which reached significance at 16–18 h after the start of cold exposure (Fig. 3d), suggesting cold resistance in these transgenic mice. After a 7-day cold exposure, AdNT3-Tg mice displayed lower fat pad mass in iWAT (Fig. 3e), enhanced expression of thermogenic genes such as *Ucp1*, carnitine palmitoyltransferase 1b, muscle (*Cpt1b*), otopetrin 1 (*Otop1*), and epithelial V-like antigen 1 (*Eva1*) in iWAT (Fig. 3f), increased UCP1 protein

expression (Fig. 3g) and appearance of more UCP1-positive multilocular cells in iWAT (Fig. 3h), indicating an increased beigeing phenotype in AdNT3-Tg mice in response to a chronic cold challenge.

Overexpressing NT-3 in adipocytes also led to increased thermogenic gene expression in iBAT, including *Ucp1*, peroxisome proliferative activated receptor gamma *(Pparγ)*, *Cpt1β*, and *Eva1* (Supplementary Fig. 5g). UCP1 protein tended to increase in iBAT of AdNT3-Tg mice after the 7-day cold exposure (Supplementary Fig. 5h). Histological analysis from both H&E staining and UCP1 immunostaining showed decreased lipid droplet size in iBAT of AdNT3-Tg mice compared to that of WT mice after cold exposure (Supplementary Fig. 5i). Our data indicate that NT-3 overexpression in adipocytes results in overall improvement of brown and beige adipocyte thermogenic function against a cold challenge.

On a regular chow diet, AdNT3-Tg did not show any differences in body weight compared to their WT littermates when measured up to 16 weeks of age (Supplementary Fig. 6a). However, when challenged with HFD, body weight of the AdNT3-Tg and WT mice started to diverge after 6–8 weeks of HFD feeding, with the AdNT3-Tg mice gaining less weight (Fig. 3i). This was associated with decreased fat pad mass in iWAT and eWAT of the transgenic mice (Fig. 3j). The reduced adiposity in the AdNT3-Tg mice might stem from increased energy expenditure, as they exhibited increased oxygen consumption and heat production (Fig. 3k, l) without changes in food intake and locomotor activity (Supplementary Fig. 6b, c). In consistence, AdNT3-Tg mice exhibited enhanced expression of BAT-specific genes such as *Ucp1*, *Cpt1b*, *Otop1*, elongation of very-long-chain fatty acids (FEN1/Elo2, SUR4/Elo3, yeast)-like 3 *(Elovl3)*, *Eva1*, and acyl-Coenzyme A oxidase 1 *(Acox1)* in iBAT (Fig. 3m), increased UCP1 and TH protein levels in iBAT as measured by immunoblotting (Fig. 3n), and smaller brown adipocytes with more UCP1 immunostaining in iBAT compared to that of WT controls (Fig. 3o). Moreover, AdNT3-Tg mice had increased expression of thermogenic genes including *Ucp1*, *Otop1*, *Eovl3*, *Cpt1b*, and *Acox1* in iWAT (Fig. 3p), and reduced adipocyte size in iWAT as shown by H&E staining compared to that of WT mice (Fig. 3q). Since the hypothalamus plays important roles in regulating brown fat thermogenesis and whole-body energy expenditure[20,26], we also measured neuropeptide expression in the arcuate hypothalamus region. We found no difference in the expression of neuropeptides pro-opiomelanocortin *(Pomc)*, neuropeptide Y *(Npy)*, and agouti-related neuropeptide *(Agrp)* in the arcuate nucleus (Supplementary Fig 6d), suggesting that obesity resistance in AdNT3-Tg mice may not be mediated through a central mechanism, at least not through a hypothalamic POMC or NPY/AGRP neuron-dependent mechanism.

We also assessed glucose homeostasis and insulin sensitivity in HFD-fed AdNT3-Tg mice. Whereas fasting glucose level was not different between AdNT3-Tg and WT mice when measured at 6, 8, and 12 weeks of HFD feeding, fasting glucose level became lower in AdNT3-Tg mice than that of WT mice after 16 weeks of HFD feeding (Supplementary Fig. 7a), at which point AdNT3-Tg mice weighed much less than their WT littermates. Similarly, there was no difference in glucose tolerance (GTT) and ITT tests between AdNT3-Tg and WT mice when measured at 6-7 weeks of HFD feeding (Supplementary Fig. 7b, c), before body weight difference was observed. However, these transgenic mice exhibited improved glucose tolerance and increased insulin sensitivity in GTT and ITT tests measured at 15–16 weeks of HFD feeding (Supplementary Fig. 7d, e), when the difference in body weight and adiposity between AdNT3-Tg and WT mice became evident. To assess whether liver gluconeogenesis could

contribute to improved glucose homeostasis in AdNT3-Tg mice, we performed a pyruvate tolerance test (PTT) in HFD-fed AdNT3-Tg and WT mice. However, we did not observe any differences in PTT between AdNT3-Tg and WT mice when measured at either 8 weeks or 14 weeks of HFD feeding (Supplementary Fig. 7f, g). Thus, our data suggest that the enhanced thermogenic program in iBAT and iWAT collectively contributes to the increased energy expenditure in transgenic mice with adipocyte overexpression of NT-3, resulting in reduced adiposity and obesity resistance. The improved glucose homeostasis and insulin sensitivity observed in AdNT3-Tg mice on HFD is most likely secondary to their obesity resistance, and their improved glucose homeostasis may not be due to changes in liver gluconeogenesis, but rather due to improved glucose utilization in the body.

To test the reproducibility of the genetic model with NT-3 overexpression, we have also generated mice that overexpress NT-3 specifically in adipocytes by inserting the *Nt-3* transgene with loxP-flanked Stop sequence into the *Rosa26* locus (Supplementary Fig. 8a, b) as previously described[27]. The resulting Rosa-NT-3$^{fl/+}$ mice were crossed with Adiponectin-Cre mice[28] (Kindly provided by Dr. Evan Rosen, Beth Israel Medical Center, Harvard Medical School) to generate mice with adipocyte-specific NT-3 knock-in mice (Adiponectin-Cre::Rosa-NT-3$^{fl/+}$, or AdNT3-KI) by deleting the loxP-flanked Stop sequence. AdNT3-KI mice had significantly increased *Nt-3* mRNA and protein levels in BAT and WAT compared to that of fl/+ controls (Supplementary Fig. 8c, d), whereas *Nt-3* expression in other tissues, including areas in the brain (hypothalamus, including ARC, VMH, PVN, and LH; cortex; and hippocampus), liver and muscle, was not different between AdNT3-KI and fl/+ mice (Supplementary Fig. 8e), indicating that *NT-3* overexpression is specific to adipose tissue. However, unlike our AdNT3-Tg mice, where *Nt-3* overexpression was primarily restricted within adipose tissue (Fig. 3a–c and Supplementary Fig. 5c–f), AdNT3-KI mice exhibited significantly increased circulating NT-3 levels compared to that of fl/+ controls (Supplementary Fig. 8f), possibly due to the high expression of *Nt-3* in adipose tissue.

Similar to AdNT3-Tg mice, AdNT3-KI mice exhibited enhanced cold-induced thermogenesis as shown by higher body temperature, lower body weight, lower fat mass, increased expression of *Ucp1* and other thermogenic genes in iWAT, higher UCP1, and TH protein levels in iWAT and more UCP1-positive beige adipocytes in iWAT than those of fl/+ control mice in response to a 7-day cold challenge (Supplementary Fig. 9a–f).

In consistence, AdNT3-KI mice had lower body weight, lower body fat content, and reduced fat pad mass in iBAT, iWAT, and rWAT than fl/+ controls when challenged with HFD (Supplementary. Fig 10a–c). The reduced adiposity in AdNT3-KI mice was primarily due to increased energy expenditure, as AdNF3-KI mice had no difference in food intake (Supplementary Fig. 10d), but had significantly increased oxygen consumption and heat production (Supplementary Fig. 10e, f), without changes in locomotor activity (Supplementary Fig. 10g). Consistent with their higher energy expenditure, AdNT3-KI mice exhibited increased *Ucp1* and other thermogenic gene expression, increased UCP1 and TH protein levels, and smaller brown adipocyte size with more intense UCP1 immunostaining in iBAT (Supplementary Fig. 10h, j).

Since we observed significantly increased serum NT-3 levels in AdNT3-KI mice, we further investigated whether the increased serum NT-3 levels could lead to pleiotropic effects of NT-3 in the brain or other peripheral tissues that may contribute to obesity resistance observed in AdNT3-KI mice. As shown in Supplementary Fig. 11, there was no difference in *Agrp*, *Pomc*, and *Npy* expression in the arcuate hypothalamus (Supplementary Fig. 11a).

In addition, there was also no difference in the expression of genes associated with mitochondria function and lipid metabolism in muscle (Supplementary Fig. 11b) and liver (Supplementary Fig. 11c) between AdNT3-KI and fl/+ mice. These data indicate that obesity resistance in AdNT3-KI mice is primarily due to increased energy expenditure stemming from increased adipose tissue thermogenesis.

We also assessed glucose homeostasis and insulin sensitivity in AdNT3-KI and WT mice fed HFD. As shown in Supplementary Fig. 12, there was no difference in GTT and ITT tests between AdNT3-KI and WT littermates when measured at 3–4 weeks of HFD feeding, respectively (Supplementary Fig. 12a, b), at which time AdNT3-KI and WT mice had similar body weight or the body weight of AdNT3-KI and WT mice just started to diverge (Supplementary Fig. 10a). However, HFD-fed AdNT3-KI mice exhibited improved glucose tolerance and insulin sensitivity in GTT and ITT tests measured at 15–16 weeks of HFD feeding (Supplementary Fig. 12c, d), when AdNT3-KI mice weighed significantly less than WT littermates (Supplementary Fig. 10a). To assess whether liver gluconeogenesis could contribute to improved glucose homeostasis in AdNT3-KI mice, we performed PTT in HFD-fed AdNT3-KI and WT mice. As shown in Supplementary Fig. 12e, there was no significant difference in PTT test between AdNT3-KI and WT mice when measured at 14 weeks of HFD feeding. Thus, our data suggest that improved glucose tolerance and insulin sensitivity in AdNT3-KI mice may be secondary to obesity resistance in these mice when fed HFD, and their improved glucose homeostasis may not be due to changes in liver gluconeogenesis, but rather due to improved glucose utilization in the body.

**NT-3 regulates sympathetic ganglia (SG) neuronal growth and activation.** We then further studied mechanisms by which adipose tissue-derived NT-3 regulates brown and beige adipocyte thermogenic function. Since NT-3 is a neurotrophic factor that regulates SNS neuron growth and target tissue innervation[24,25], we first tested NT-3's effect on sympathetic neurite growth in primarily cultured sympathetic neurons in vitro. We found that NT-3 (100 ng/ml) treatment significantly stimulated sympathetic neuron neurite growth, as shown by the increased number of neurites per neuron, total neurite length, and maximal neurite length per neuron, as well as average neurite length per neurite per neuron (Fig. 4a).

The overexpression of NT-3 in our AdNT3-Tg model is primarily restricted to the adipose tissue, without any confounding effects on the changes in the circulating NT-3. Thus, to further assess the effect of adipose tissue NT-3 on sympathetic nerve innervation into adipose tissue in vivo, we immunostained sympathetic nerves innervating adipose tissue with antibodies against sympathetic marker TH using the Adipo-Clear approach[22] in AdNT3-Tg mice. Analysis of the sympathetic nerve fibers by Imaris Image Analysis Software revealed a significantly higher mean nerve fiber length and mean nerve fiber branching points in iWAT of 2-month-old AdNT3-Tg mice than that in WT mice when housed at ambient room temperature (Supplementary Fig. 13). In addition, a 7-day cold exposure also increased mean nerve fiber length and nerve fiber branching points in eWAT and iWAT of 2-month-old AdNT3-Tg mice compared to that of WT mice (Fig. 4b, c). Our data thus support an important role of adipose tissue-derived NT-3 in regulating sympathetic nerve innervation in fat tissues, which may contribute to its effects on promoting brown and beige adipocyte thermogenic function during cold and diet challenges as observed in NT-3-injected, AdNT3-Tg, and AdNT3-KI mice (Figs. 2–3 and Supplementary Figs. 5–12).

**NT-3/TRKC regulates sympathetic innervation in adipose tissue.** NT-3 binds with high affinity to its cognitive receptor, neurotrophic receptor 3/Tropomyosin receptor kinase C (NTRK3/TRKC)[29–31]. TRKC immunoreactivity has been found in the majority of sympathetic neurons in SG during embryonic development and in adult animals[32,33], indicating the importance of TRKC signaling in regulating sympathetic neuron growth and development.

To further explore mechanisms by which NT-3 regulates sympathetic innervation in adipose tissue, we investigated whether TRKC, the receptor for NT-3, is important in regulating sympathetic innervation in adipose tissue. Homozygous TRKC −/− mice die by postnatal day 21[34]. Thus, we studied mice with haploinsufficiency of TRKC (TRKC + /−). In addition, we also generated mice with sympathetic-specific deletion of TRKC by crossing the TRKC-floxed mouse model (fl/fl) where the TRKC kinase ATP-binding domain is flanked with loxP sites[35] with TH-Cre mice where Cre-recombinase expression is under the control of the TH promoter[36] to achieve specific deletion of TRKC in sympathetic neurons (TH-Cre::TRKC-fl/fl, or STRKCKO). As expected, *Trkc* expression in SG was reduced to ~50% in TRKC + /− mice compared to that of WT littermates (Supplementary Fig. 14a). In addition, we found that *Trkc* expression was reduced around 75% in SG of STRKCKO mice (Supplementary Fig. 14b).

Interestingly, sympathetic neurons isolated from TRKC + /− mice had significantly suppressed neuronal growth, as shown by decreased total neurite length and maximal neurite length per neuron (Fig. 5a). To see whether NT-3's neurotrophic effects require TRKC, we treated sympathetic neurons isolated from fl/fl or STRKCKO mice with NT-3 and found that deleting TRKC in sympathetic neurons significantly suppressed basal- and NT-3-induced neurite growth (Fig. 5b), indicating that TRKC is required for NT-3's neurotrophic effects.

Using Adipo-Clear approach with TH immunostaining, we found that TRKC + /− mice had significantly less sympathetic innervation in adipose tissue in response to a 7-day cold challenge when compared to that of WT control mice, as shown by reduced mean nerve fiber length and mean nerve fiber branching points in eWAT and iWAT of TRKC + /− (Fig. 5c, d). In addition, sympathetic neuron-specific deletion of TRKC in STRKCKO mice also led to significantly reduced mean sympathetic nerve fiber length and mean sympathetic nerve fiber branching point in eWAT and iWAT compared to that of fl/fl littermates during a chronic 7-day cold challenge (Fig. 5e, f). Thus, our data strongly support the importance of NT-3/TRKC in regulating sympathetic innervation in adipose tissue.

**NT-3/TRKC regulates SG neuronal activity.** To further study whether NT-3/TRKC regulates the activity of SG that specifically innervates iBAT, we injected fluorescent retrograde neuronal tracer Fast Blue (FB) into iBAT of TRKC-Cre::R26-stop-EYFP reporter mice. Animals were given 10–14 days to recover and allow for retrograde FB transport and were then subjected to a 7-day cold challenge. SG at the thoracic T2 level, which innervates iBAT[37], were analyzed for single-, double-, and triple-labeled neurons (Fig. 6a–c). Interestingly, the majority of the TRKC-positive neurons in T2 SG were colocalized with TH (TH + TRKC +) (Fig. 6a, b). In addition, ~17% of TH + TRKC + neurons in mice housed at room temperature were also colocalized with FB, indicating that these neurons specifically innervate iBAT (Fig. 6c), with the remaining 83% of FB- neurons in TH + TRKC + populations in the T2 SG innervate organs other than iBAT (Fig. 6c). Cold exposure significantly increased the percentage of TH + TRKC + FB + neurons in T2 ganglia (Fig. 6b,

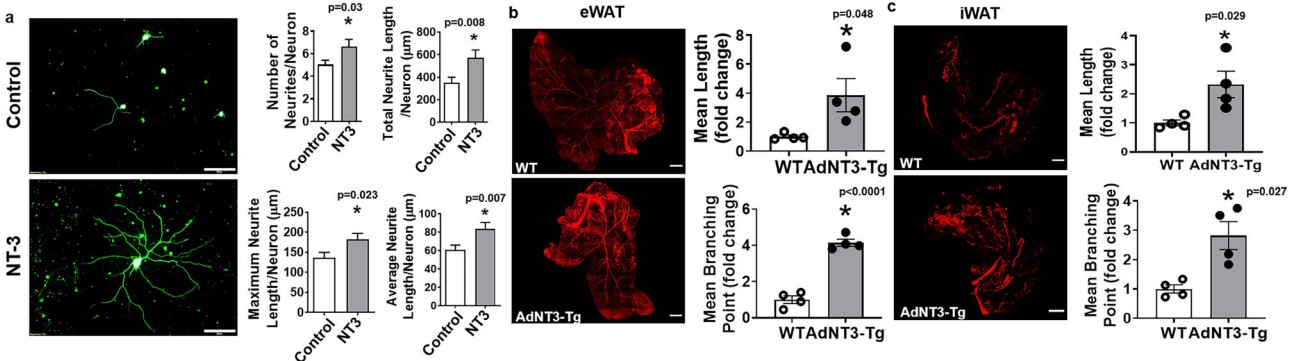

**Fig. 4 NT-3 regulates sympathetic neuronal growth and innervation in adipose tissue. a** Representative βIII-tubulin immunofluorescent images of sympathetic ganglia neurons treated with NT-3 (100 ng/ml) (left panel, from three independent treatments/group, scale bar = 100 μm) and quantitation of the number of neurites per neuron, total neurite length per neuron, maximal neurite length per neuron, and average neurite length per neurite per neuron (right panel, control $n = 80$, NT-3 $n = 60$, * indicates statistical significance between control and NT-3 treatments with unpaired two-tailed $t$ test). Neurons from sympathetic ganglia at thoracic T1–T4 levels from 6- to 8-week-old mice were isolated and used in the experiment. **b, c** Representative images of TH-positive sympathetic nerve innervation in eWAT (**b**, left panel, from four replicate animals/group, scale bar = 2000 μm) and iWAT (**c**, left panel, from four replicate animals/group, scale bar = 2000 μm) and quantitation of mean nerve fiber length and mean nerve fiber branching points normalized to total adipose tissue area in eWAT (**b**, right panel, $n = 4$/group, * indicates statistical significance between WT and AdNT3-Tg with unpaired two-tailed $t$ test) and iWAT (**c**, right panel, $n = 4$/group, * indicates statistical significance between WT and AdNT3-Tg with unpaired two-tailed $t$ test) in AdNT3-Tg mice and their littermate control mice after a 7-day cold challenge. All data are expressed as mean ± SEM.

c), with concomitant decreases in the percentage of TH+ single-labeled and TH+ FB+ double-labeled neurons; whereas there was no significant change in the percentage of TH+ TRKC+ neurons that innervate other organs (TH+ TRKC+ FB-) (Fig. 6b, c). Our data indicate that cold exposure specifically increases TH+ TRKC+ neurons in the SG that innervate brown fat (TH+ TRKC+ FB+ neurons), whereas it does not affect TH+ TRKC+ neurons that might innervate organs other than iBAT (TH+ TRKC+ FB- neurons). The decrease of TH+ single-labeled and TH+ FB+ double-labeled neurons with a concomitant increase of TH+ TRKC+ FB+ neurons in response to cold is intriguing. It is possible that cold exposure stimulates the expression of TRKC in those TH+ single-labeled and TH+ FB+ double-labeled neurons. Since NT-3/TRKC signaling stimulates neurite growth (Figs. 4 and 5), this could result in increased axonal growth, leading to increased innervation specifically to adipose tissue from those neurons that now express TRKC (i.e., TH+ TRKC+ FB+ neurons).

We then injected FB into iBAT of AdNT3-Tg, TRKC+/−, and their respective control mice and subjected them to a 7-day cold challenge. SG at thoracic T2 level were collected for immunohistochemistry analysis with antibodies against TH as the sympathetic marker or cFos as the neuronal activation marker, and double-stained with FB. Interestingly, overexpressing NT-3 in adipose tissue significantly increased (Fig. 6d, e), while TRKC haploinsufficiency significantly decreased (Fig. 6f, g) TH/FB or cFos/FB double-labeled neurons in T2 SG during cold exposure.

We have similarly injected FB into iWAT of AdNT3-Tg, TRKC+/−, and their respective littermate control mice and subjected them to a 7-day cold challenge. SG at lumbar L1 level, which innervate iWAT[37], were collected for immunohistochemistry analysis for TH/FB or cFos/FB immunostaining. We found that TH/FB or cFos/FB double-labeled neurons were significantly increased in L1 SG of AdNT3-Tg mice; but significantly decreased in L1 SG of TRKC+/− mice when compared to their respective littermate controls (Supplementary Fig. 15a, b).

Consistent with reduced sympathetic innervation and activity in TRKC+/− mice, norepinephrine (NE) level was reduced in eWAT and tended to reduce in iWAT after a 16-h cold exposure in TRKC+/− mice (Supplementary Fig. 16a). In addition, NE

turnover rate (NETO) was also significantly reduced in all fat tissues including iBAT, iWAT, eWAT, and rWAT after cold exposure in TRKC+/− mice (Supplementary Fig. 16b). Since TRKC+/− mice had haploinsufficiency of TRKC in all neurons, which might affect sympathetic innervation in other tissues as well, we also measured NE content and NETO in other tissues, including skeletal muscle, kidney, and heart. However, we found there was no difference in either NE content or NETO in these tissues between WT and TRKC+/− mice after a 16-hour cold exposure (Supplementary Fig. 16c, d).

Thus, our data strongly support the importance of adipose tissue-derived NT-3 and TRKC in the regulation of peripheral sympathetic neuron activity in SG that directly innervate brown and white adipose tissue.

**NT-3/TRKC regulates a plethora of transcriptional program that controls neuronal axonal growth and elongation in SG.** To gain further insight into how NT-3/TRKC regulates the sympathetic neuronal function and axonal growth, we performed RNA-seq analysis in thoracic T1–T4 SG that innervate iBAT[37] from 1-day cold-challenged AdNT3-Tg, TRKC+/− and their respective WT control mice.

Our RNA-seq analysis demonstrated that pathways involved in neuronal axonal growth were regulated in an opposite direction in SG of AdNT3-Tg and TRKC+/− mice (Fig. 7a). For example, axonal outgrowth and guidance to the proper target tissues require the coordination of cellular programs, including the coordination of actin filaments and microtubules, the dynamic cytoskeletal polymers, and "building blocks" that promote shape change and locomotion during the peripheral nerve axonal outgrowth and elongation[38,39]. Interestingly, we found several myosins, actions, and troponins were upregulated in AdNT3-Tg mice while downregulated in TRKC+/− mice, including myosin heavy polypeptide 1 (Myh1), myosin heavy polypeptide 4 (Myh4), actin alpha 1 (Acta1), troponin C2 (Tnnc2), troponin I skeletal, fast 2 (Tnni2), troponin T2 (Tnnt2), muscle creatine kinase (Ckm), and sarcoplasmic/endoplasmic reticulum calcium ATPase 1 (Atp2a1) (Fig. 7a). Although these genes are typically characteristic of the striated muscle contraction pathway, similar genes in this pathway, including Myh1 and ATP2a1, were shown to be decreased in the brain of the neuron degenerative disease

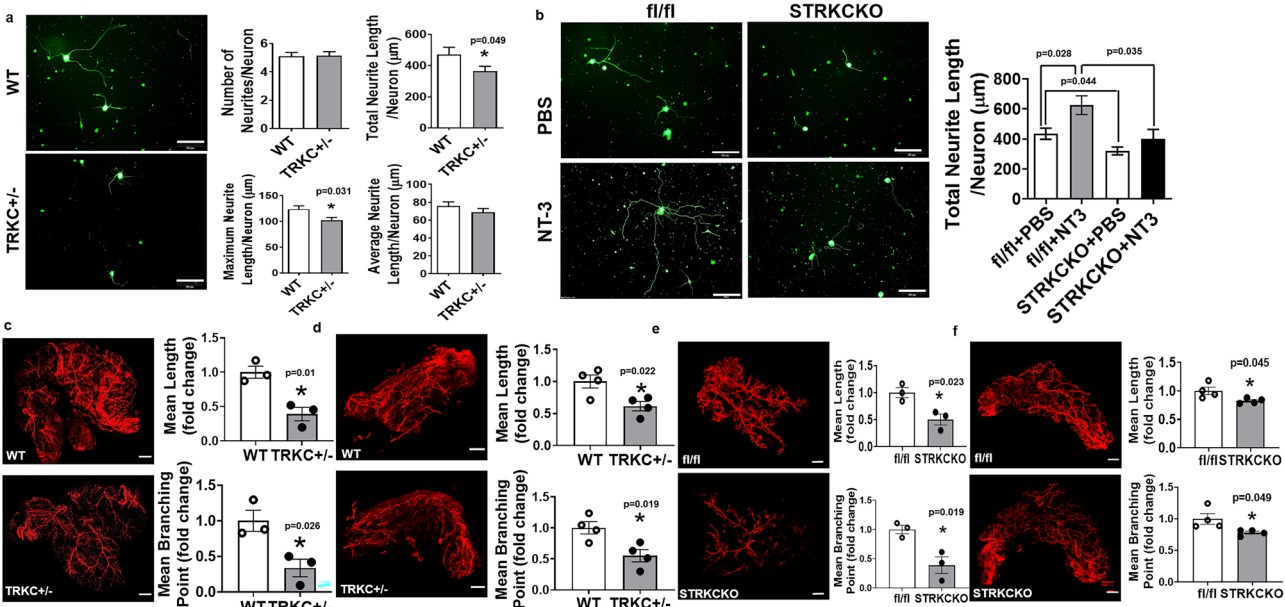

**Fig. 5 TRKC is required for the neurotrophic effects of NT-3. a** Representative βIII-tubulin immunofluorescent images of sympathetic ganglia neurons isolated from WT and TRKC + /− mice (left panel, from three independent treatments/group, scale bar = 100 μm) and quantitation of the number of neurites per neuron, total neurite length per neuron, maximal neurite length per neuron, and average neurite length per neurite per neuron (right panel, control n = 158, TRKC + /− n = 162, * indicates statistical significance between WT and TRKC + /− with unpaired two-tailed t test). **b** Representative βIII-tubulin immunofluorescent images of sympathetic ganglia neurons isolated from fl/fl and STRKCKO mice treated with phosphate-buffered saline (PBS) or NT-3 (100 ng/ml) (left panel, from three independent treatments/group, scale bar = 100 μm) and quantitation of total neurite length per neuron (right panel, fl/fl n = 135, fl/fl + NT-3 n = 49, STRKCKO n = 168, STRKCKO + NT-3 n = 42. Statistical significance was analyzed with two-way ANOVA. fl/fl vs STRKCKO $F$ (1, 390) = 13.25 $P$ = 0.0003; PBS vs NT-3 $F$ (1, 390) = 7.680 $P$ = 0.0059. * Indicates statistical significance with Turkey's multiple comparisons test). Neurons from sympathetic ganglia at thoracic T1–T4 levels from 6- to 8-week-old mice were isolated and used in these experiments. **c, d** Representative images of TH-positive sympathetic nerve innervation in eWAT (**c**, left panel, from three replicate animals/group, scale bar = 2000 μm) and iWAT (**d**, left panel, from four replicate animals/group, scale bar = 2000 μm) and quantitation of mean nerve fiber length and mean nerve fiber branching points normalized to total adipose tissue area in eWAT (**c**, right panel, n = 3/group, * indicates statistical significance between WT and TRKC + /− with unpaired two-tailed t test) and iWAT (**d**, right panel, n = 4/group, * indicates statistical significance between WT and TRKC + /− with unpaired two-tailed t test) in TRKC + /− mice and their littermate control mice after a 7-day cold challenge. **e, f** Representative images of TH-positive sympathetic nerve innervation in eWAT (**e**, left panel, from three replicate animals/group, scale bar = 2000 μm) and iWAT (**f**, left panel, from four replicate animals/group, scale bar = 2000 μm) and quantitation of mean nerve fiber length and mean nerve fiber branching points normalized to total adipose tissue area in eWAT (**e**, right panel, n = 3/group, * indicates statistical significance between fl/fl and STRKCKO with unpaired two-tailed t test) and iWAT (**f**, right panel, n = 4/group, * indicates statistical significance between fl/fl and STRKCKO with unpaired two-tailed t test) in fl/fl and STRKCKO mice after a 7-day cold challenge. All data are expressed as mean ± SEM.

Amyotrophic lateral sclerosis (ALS)[40]. Thus, these myosin and actin filament components may be previously unidentified members of the cytoskeletal family regulated by NT-3/TRKC signaling and are important for axonal growth and elongation in SG.

In addition, recent evidence also supports the axonal transport and translation of specific mRNAs as a mechanism of quickly localizing needed proteins at the axons and axonal growth cones[38,39]. For example, β-actin mRNA was localized to axons and axonal growth cones, and this localization can be induced by NT-3 signaling; blocking this process also blocked NT-3-induced protein localization at the growth cone and decreased growth cone motility[41,42]. Here, we found that the serine/threonine kinase U2AF homology motif (UHM) kinase 1 (*Uhmk1*) and 5'–3' exonuclease 1 (*Xrn1*), genes that are involved in synaptic mRNA transport and translation[43,44], were upregulated in AdNT3-Tg mice but downregulated in TRKC + /− mice (Fig. 7a).

Further, the multifunctional adaptor protein casitas B-lineage lymphoma (*Cbl*) has been shown to be necessary for the recruitment of NGF/TRKA to the lipid raft and mediates signals important for actin reorganization and neurite growth[45,46]. Interestingly, we found that *Cbl* was upregulated in AdNT3-Tg

mice but downregulated in TRKC + /− mice (Fig. 7a), indicating *Cbl* may also be regulated by NT-3/TRKC signaling.

Additional genes that were oppositely regulated in AdNT3-Tg and TRKC + /− mice include proteins that regulate cytoskeletal dynamics and activities (MX dynamin-like GTPase 1 (*Mx1*), transglutaminase 1, k polypeptide (*Tgm1*), and NHS actin remodeling regulator (*Nhs*)); proteins involved in glucose metabolism (phosphoglucomutase 1 (*Pgm1*); intracellular protein sorting (thyroid hormone receptor interactor 11 (*Trip11*), mind-bomb E3 ubiquitin-protein ligase 1 (*Mib1*)) and the chemokine (c–x–c motif) ligand 10 (*Cxcl10*) (Fig. 7a).

Using real-time RT-PCR approaches, we have validated some of the reciprocally regulated genes in AdNT3-Tg and TRKC + /− mice, including *Cxcl10* (Fig. 7b, c). Interestingly, it was reported that *Cxcl10* in the spinal cord was upregulated following spinal cord injury, which was critical in recruiting T lymphocytes and inducing apoptosis at the injury site, whereas neutralizing CXCL10 resulted in reduced apoptosis and increased axonal sprouting following spinal cord injury[47,48]. To further study whether CXCL10 might be a potential link between neuronal inflammation and the suppression of neuronal axonal growth, we treated primary sympathetic neurons with CXCL10. Interestingly, we found that CXCL10 (100 ng/ml) significantly reduced basal-

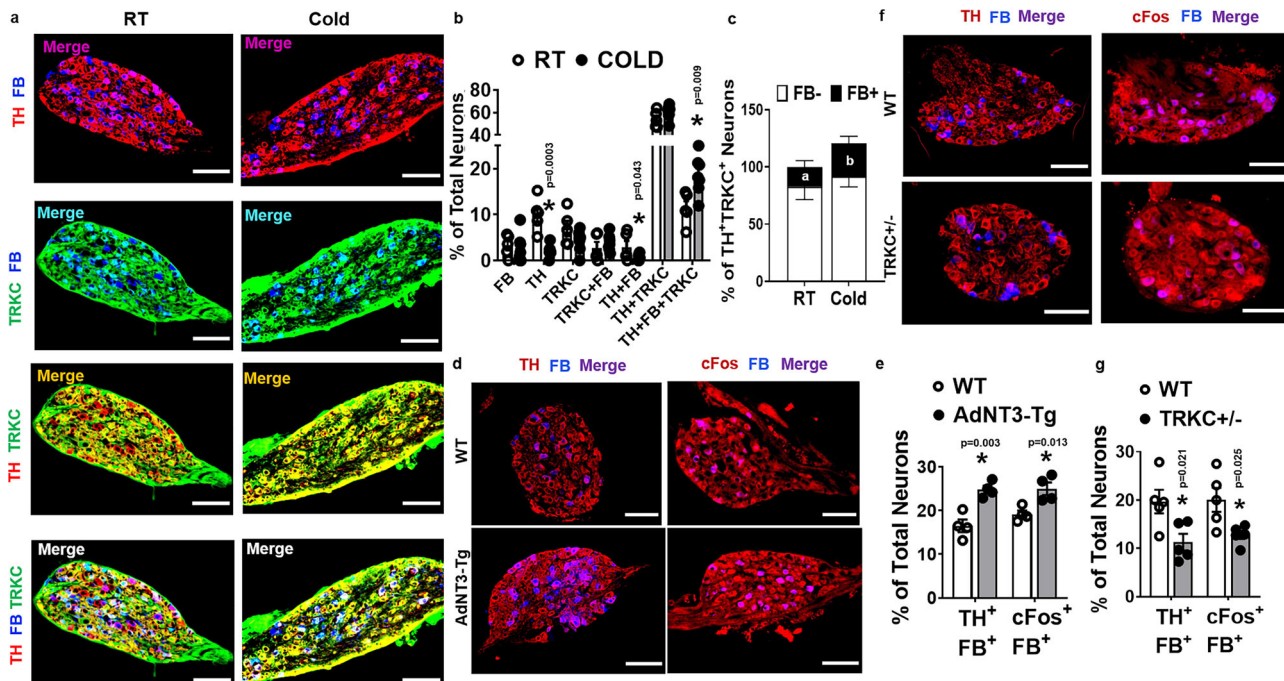

**Fig. 6 NT-3/TRKC regulates sympathetic ganglia neuronal activity. a, b** Representative images of TH, Fast Blue (FB) and TRKC-EYFP labeling (**a**, from five replicate animals for RT, and 7 replicate animals for cold, scale bar = 75 μm), and percentage of single-, double-, and triple-labeled neurons (**b**, RT = 5, cold = 7, * indicates statistical significance between RT and cold treatments with unpaired two-tailed *t* test) in sympathetic ganglia at thoracic T2 level in TRKC-Cre::R26EYFP mice at room temperature (RT) or subjected to a 7-day cold challenge. In panel **b**, the calculation of single-, double-, and triple-labeled neurons is mutually exclusive, i.e., the number of single-labeled neurons does not include double- or triple-labeled neuron numbers, and the number of double-labeled neurons does not include triple-labeled neuron numbers. **c** Percentage of FB-positive (FB⁺) and FB-negative (FB⁻) neurons in TH/TRKC-positive neurons in sympathetic ganglia at thoracic T2 level in TRKC-Cre::R26EYFP mice at room temperature (RT) or subjected to a 7-day cold challenge (RT = 5, COLD = 7, a vs b: FB⁺ in RT vs FB⁺ in cold, *P* = 0.010 with unpaired two-tailed *t* test). **d, e** Representative images (**d**, from four replicate animals/group, scale bar = 75 μm) and quantitation (**e**, *n* = 4/group, * indicates statistical significance between WT and AdNT3-Tg with unpaired two-tailed *t* test) of TH/FB and cFos/FB labeling in sympathetic ganglia at thoracic T2 level in WT and AdNT3-Tg mice after a 7-day cold challenge. **f, g** Representative images (**f**, from five replicate animals/group, scale bar = 75 μm) and quantitation (**g**, *n* = 5/group, * indicates statistical significance between WT and TRKC + /− with unpaired two-tailed *t* test) of TH/FB and cFos/FB labeling in sympathetic ganglia at thoracic T2 level in WT and TRKC + /− mice after a 7-day cold challenge. All data are expressed as mean ± SEM.

and NT-3-stimulated neurite growth in cultured sympathetic neurons (Fig. 7d). Thus, our data have identified a potential mechanism linking inflammation to the suppression of neuronal axonal growth.

**Mice with haploinsufficiency of TRKC exhibit impaired cold-induced thermogenesis and are prone to diet-induced obesity.** We then further studied the importance of TRKC in regulating cold- and diet-induced thermogenesis and whole-body energy homeostasis. During a 7-day cold challenge, TRKC + /− mice exhibited lower body temperature (Fig. 8a) and were therefore more cold-sensitive than their littermate controls. During the cold challenge, TRKC + /− mice had significantly reduced *Ucp1* and other thermogenic gene expression in iWAT, retroperitoneal WAT (rWAT) and iBAT (Fig. 8b–d), reduced UCP1 and TH protein levels in iWAT and rWAT (Fig. 8e, f), and larger adipocytes but much fewer UCP1-positive multilocular beige adipocytes in iWAT and rWAT (Fig. 8g). However, there was no significant difference in UCP1 protein levels and UCP1 immunostaining in iBAT between TRKC + /− and WT control mice during the 7-day chronic cold challenge (Supplementary Fig. 17a, b). This is possibly because of the high endogenous expression *Nt-3* and thus NT-3/TRKC signaling in iBAT as compared to that of other WAT depots, thus, sufficient sympathetic innervation could still be maintained even with the deletion of one allele of TRKC in TRKC + /− mice. Alternatively, our data may suggest

that TRKC is especially important in regulating cold-induced thermogenesis and beigeing in WAT.

We also studied the metabolic phenotypes in TRKC + /− mice fed with either regular chow or HFD. There was no difference in body weight between TRKC + /− and WT littermates fed a regular chow diet when measured up to 16 weeks of age (Supplementary Fig. 18). However, when housed at ambient room temperature (20–22 °C), TRKC + /− mice on HFD had increased fat mass in eWAT and rWAT and had a tendency of increase in iWAT mass, despite no change in body weight and food intake (Supplementary Fig. 19a–c). Interestingly, TRKC + /− mice exhibited decreased energy expenditure evident by reduced oxygen consumption and heat production (Supplementary Fig. 19d, e) without changes in locomotor activity (Supplementary Fig. 19f). The decreased energy expenditure in TRKC + /− mice was associated with reduced UCP1 and TH protein levels in iBAT (Supplementary Fig. 19g). Mice with haploinsufficiency of TRKC also showed glucose intolerance and insulin resistance in GTT and ITT tests, respectively (Supplementary Fig. 19h, i).

We also performed HFD feeding in these animals under thermoneutrality (30 °C) to avoid any nonshivering thermogenesis that may be induced by mild cold stress in mice housed under ambient room temperature (20–22 °C) as we recently reported[49], which may serve as a confounding factor in assessing diet-induced thermogenesis[4,50,51]. When housed at thermoneutrality, HFD-fed TRKC + /− mice consistently gained more weight

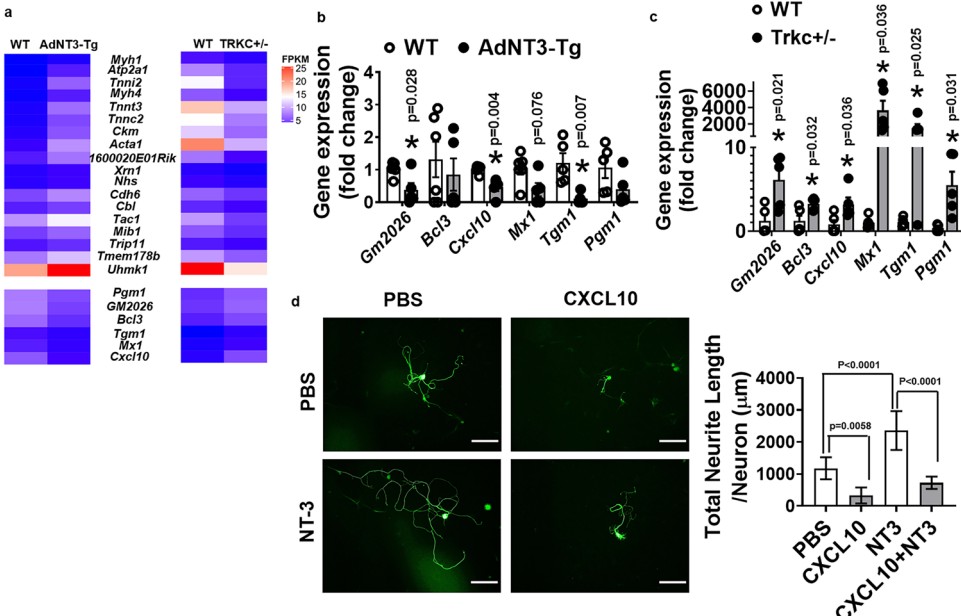

**Fig. 7 RNA-seq analysis of sympathetic ganglia from AdNT3-Tg and TRKC + /− mice after 1-day cold challenge. a** Heatmap of the expression of genes regulated in the opposite direction in sympathetic ganglia from thoracic T1 to T4 levels of AdNT3-Tg and TRKC + /− mice after 1-day cold challenge. T1–T4 sympathetic ganglia from six mice each group was pooled for RNA-seq analysis. **b, c** Real-time RT-PCR validation of the genes that are downregulated in sympathetic ganglia of AdNT3-Tg mice (**b**, $n = 5$ for WT (except for Tgm1 and Pgm1, where $n = 4$); $n = 5$ for AdNT3-Tg. * Indicates statistical significance between WT and AdNT3-Tg with unpaired two-tailed $t$ test), but are upregulated in sympathetic ganglia of TRKC + /− mice (**c**, $n = 5$/group, * indicates statistical significance between WT and TRKC + /− with unpaired two-tailed $t$ test) after 1-day cold exposure. **d** Measurement of neurite growth in sympathetic neurons treated with NT-3 (100 ng/ml) in the presence or absence of CXCL10 (100 ng/ml) (PBS $n = 50$, CXCL10 $n = 43$, NT-3 $n = 47$, CXCL10 + NT-3 $n = 40$. Statistical significance was analyzed by two-way ANOVA; with or without CXCL10 treatment, $F (1, 210) = 95.47$ $P < 0.0001$; with or without NT-3 treatment, $F (1, 210) = 35.62$ $P < 0.0001$. * Indicates statistical significance as shown in (**d**) with Turkey's multiple comparisons test. Scale bar = 100 μm). Neurons from sympathetic ganglia at thoracic T1–T4 levels were isolated and used in this experiment. All data are expressed as mean ± SEM.

(Fig. 8h) and had increased fat mass in iBAT, iWAT, and eWAT (Fig. 8i) with larger adipocytes sizes (Supplementary Fig. 20a) compared to their WT control mice. Moreover, TRKC + /− mice had downregulated expression of thermogenic genes in iBAT (Fig. 8j) and had a tendency of reduced UCP1 protein levels and significantly reduced TH protein levels in iBAT (Fig. 8k). This was associated with less UCP1 immunostaining and enlarged brown adipocytes in iBAT (Fig. 8l). TRKC + /− mice also exhibited reduced thermogenic gene expression in both iWAT and rWAT (Supplementary Fig. 20b, c). Consistent with increased adiposity, TRKC + /− mice exhibited glucose intolerance and insulin resistance in GTT and ITT tests, respectively (Supplementary Fig. 20d, e).

**Mice with specific deletion of TRKC in sympathetic nerves exhibit impaired cold-induced thermogenesis and are prone to diet-induced obesity.** To more specifically study the role of sympathetic neuron TRKC in regulating energy homeostasis, we generated mice with sympathetic neuron-specific deletion of TRKC (STRKCKO) (please see Supplementary Fig. 14b). When challenged with a cold exposure, STRKCKO mice had significantly reduced body temperature in response to cold exposure (Fig. 9a), but had a higher fat mass in iWAT and eWAT depots (Fig. 9b) compared to fl/fl mice. STRKCKO mice exhibited reduced *Ucp1* and other thermogenic gene expression (Fig. 9c) and larger brown adipocytes with reduced UCP1 immunostaining in iBAT during cold exposure (Fig. 9d). Further, STRKCKO mice had reduced *Ucp1* and other thermogenic gene expression (Fig. 9e) and reduced UCP1-positive beige adipocytes in iWAT in response to the 7-day cold challenge (Fig. 9f). These data indicate

that sympathetic TRKC is important in regulating cold-induced brown and beige adipocyte thermogenesis.

We then put STRKCKO mice on HFD at ambient room temperature. Body composition analysis by Minispec NMR revealed an increased fat mass with decreased lean mass in STRKCKO mice, albeit there was no change of body weight (Supplementary Fig. 21a–c). Similar to TRKC + /− mice, STRKCKO mice exhibited no change in food intake (Supplementary Fig. 21d). However, STRKCKO mice displayed decreased energy expenditure as shown by decreased oxygen consumption and heat production (Supplementary Fig. 21e, f), with no change in locomotor activity (Supplementary Fig. 21g). The reduced energy expenditure in STRKCKO mice was associated with significantly reduced UCP1 and TH protein levels in iBAT, as well as larger brown adipocytes with reduced UCP1 immunostaining in iBAT (Supplementary Fig. 21h, i). STRKCKO mice also had reduced thermogenic gene expression and larger adipocytes in iWAT (Supplementary Fig. 21j, k). Consistent with their increased adiposity, these mice showed glucose intolerance and insulin resistance in GTT and ITT tests, respectively (Supplementary Fig. 21l, m).

We also performed HFD feeding in these animals under thermoneutrality (30 °C) to better assess diet-induced thermogenesis[4,50,51]. STRKCKO mice similarly displayed higher fat mass and lower lean mass compared to their control mice under thermoneutrality (Fig. 9g) despite no change in body weight. Histological analysis also revealed that STRKCKO mice had enlarged adipocyte size in both iBAT and iWAT (Fig. 9h), indicating increased lipid accumulation. In consistence, STRKCKO mice had significantly reduced UCP1 and TH protein levels in iBAT with reduced UCP1 immunostaining and larger

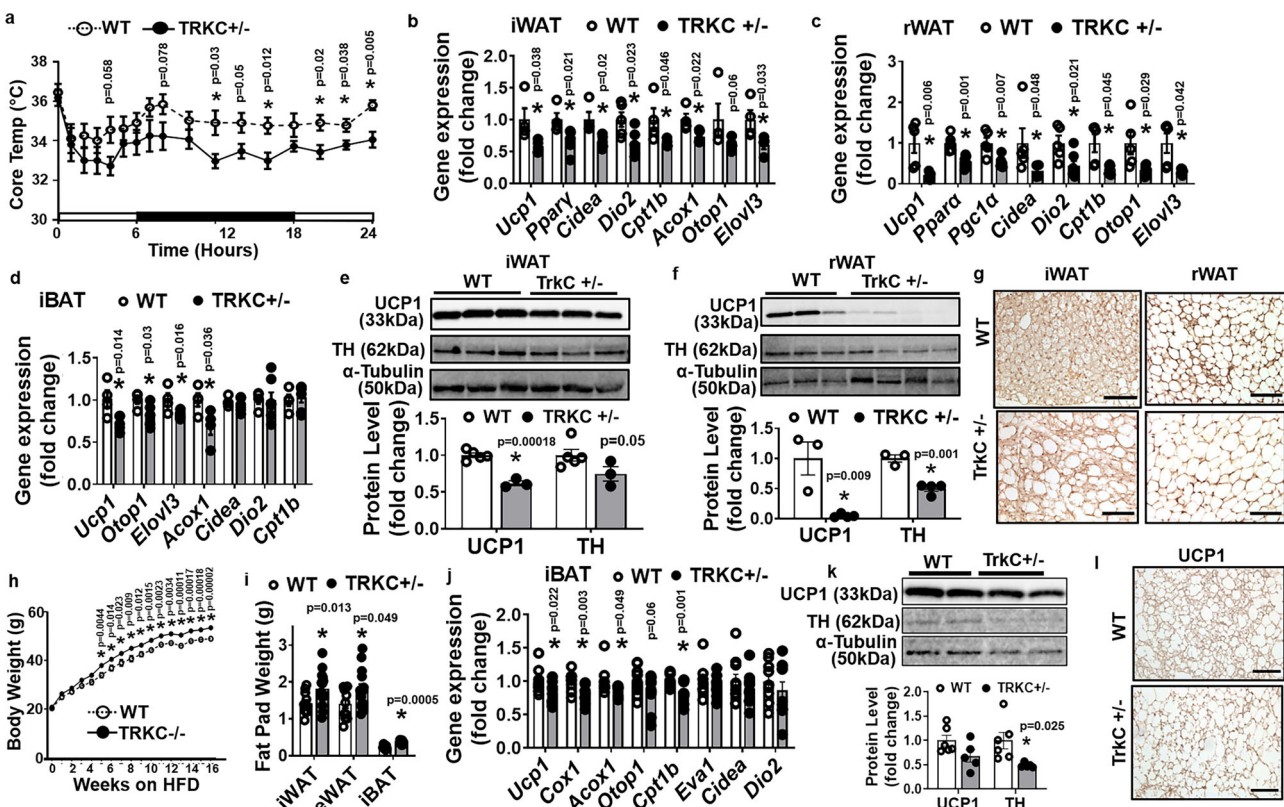

**Fig. 8 Mice with haploinsufficiency of TRKC exhibit impaired cold-induced thermogenesis and are prone to diet-induced obesity. a–d** Body temperature (**a**, WT $n = 8$, TRKC + /− $n = 6$, * indicates statistical significance between WT and TRKC + /− with unpaired two-tailed $t$ test), and Ucp1 and other thermogenic gene expression in iWAT (**b**, WT $n = 4$, TRKC + /− $n = 5$), rWAT (**c**, WT $n = 5$, TRKC + /− $n = 6$) and iBAT(**d**, WT $n = 4$, TRKC + /− $n = 5$) of WT and TRKC + /− mice during a 7-day cold exposure (* indicates statistical significance between WT and TRKC + /− with unpaired two-tailed $t$ test). **e–g** UCP1 and TH protein levels in iWAT (**e**, WT $n = 5$, TRKC + /− $n = 3$, * indicates statistical significance between WT and TRKC + /− with unpaired two-tailed $t$ test) and rWAT (**f**, WT $n = 3$, TRKC + /− $n = 4$, * indicates statistical significance between WT and TRKC + /− with unpaired two-tailed $t$ test), and representative UCP1 immunostaining images showing UCP1-positive beige adipocytes in iWAT and rWAT (**g**, from three replicate animals/group, scale bar = 150 μm) of WT and TRKC + /− mice during a 7-day cold exposure. **h–l** Body weight (**h**, WT $n = 7$, TRKC + /− $n = 8$, * indicates statistical significance between WT and TRKC + /− with unpaired two-tailed $t$ test), Fat pad mass (**i**, iBAT, WT $n = 7$, TRKC + /− $n = 8$; iWAT and eWAT, WT $n = 11$, TRKC + /− $n = 15$, *indicates statistical significance between WT and TRKC + /− with unpaired two-tailed $t$ test), Ucp1 and other thermogenic gene expression in iBAT (**j**) (BAT WT $n = 9$, TRKC + /− $n = 10$, * indicates statistical significance between WT and TRKC + /- with unpaired two-tailed $t$ test), UCP1 and TH protein levels in iBAT (**k**) (WT $n = 6$, TRKC + /− $n = 5$, *indicates statistical significance WT and TRKC + /− with unpaired two-tailed $t$ test), and representative UCP1 immunostaining in iBAT (**l**, from three replicate animals/group, scale bar = 75 μm) of WT and TRKC + /− mice fed a HFD under thermoneutrality (30 °C). All data are expressed as mean ± SEM.

adipocytes (Fig. 9i, j). With increased adiposity, STRKCKO mice exhibited glucose intolerance and insulin resistance in GTT and ITT tests, respectively (Fig. 9k, l). In sum, our data suggest that sympathetic TRKC signaling is crucial in regulating cold- and diet-induced thermogenesis and whole-body energy homeostasis.

## Discussion

The premise of this study was derived from several prior observations. For one, Xue et al. discovered the developmentally induced beige adipocytes that appear transiently, peaking at postnatal day 20 and then disappearing thereafter toward adulthood[11]. However, the fundamental question regarding the mechanisms underlying the induction and disappearance of the developmental beige adipocytes remains unanswered. Moreover, several lines of evidence have demonstrated a key role of SNS in BAT/beige thermogenesis[12–16]. Factors that promote sympathetic innervation into BAT and WAT in response to developmental environmental cues (e.g., cold) are understudied. Recent studies indicate that HFD results in significantly reduced sympathetic innervation in adipose tissue[52], suggesting that the decline of sympathetic innervation in adipose tissue under physiological

conditions is not uncommon. The HFD-induced decline in adipose tissue sympathetic innervation is mediated via a central leptin-BDNF regulated pathway that involves POMC and AGRP neurons in the ARC hypothalamus and BDNF neurons in the PVN hypothalamus[52]. Here, we discovered an important peripheral fat-derived neurotrophic factor NT-3 and its receptor TRKC in the sympathetic ganglia as key regulators of SNS growth and innervation in adipose tissue.

CNS-originated activation of SNS that drives brown and beige fat thermogenesis has been a major focus of recent investigation[20]. However, prior studies have also demonstrated peripherally derived neurotrophic factors in SNS growth and activation. The neurotrophic factors or neurotrophins, including NGF, BDNF, NT-3 and NT4/5, are a family of closely related proteins that act as survival factors for sympathetic and sensory neurons, and control the survival, development, and function of neurons in both the CNS and peripheral nervous system (PNS)[29,30,53]. In peripheral nerve systems including SNS, neurotrophins are usually synthesized at a considerable distance from neurons by nonneuronal cells (targets) that are contacted by axons of these neurons innervating the target tissue[29,30,53].

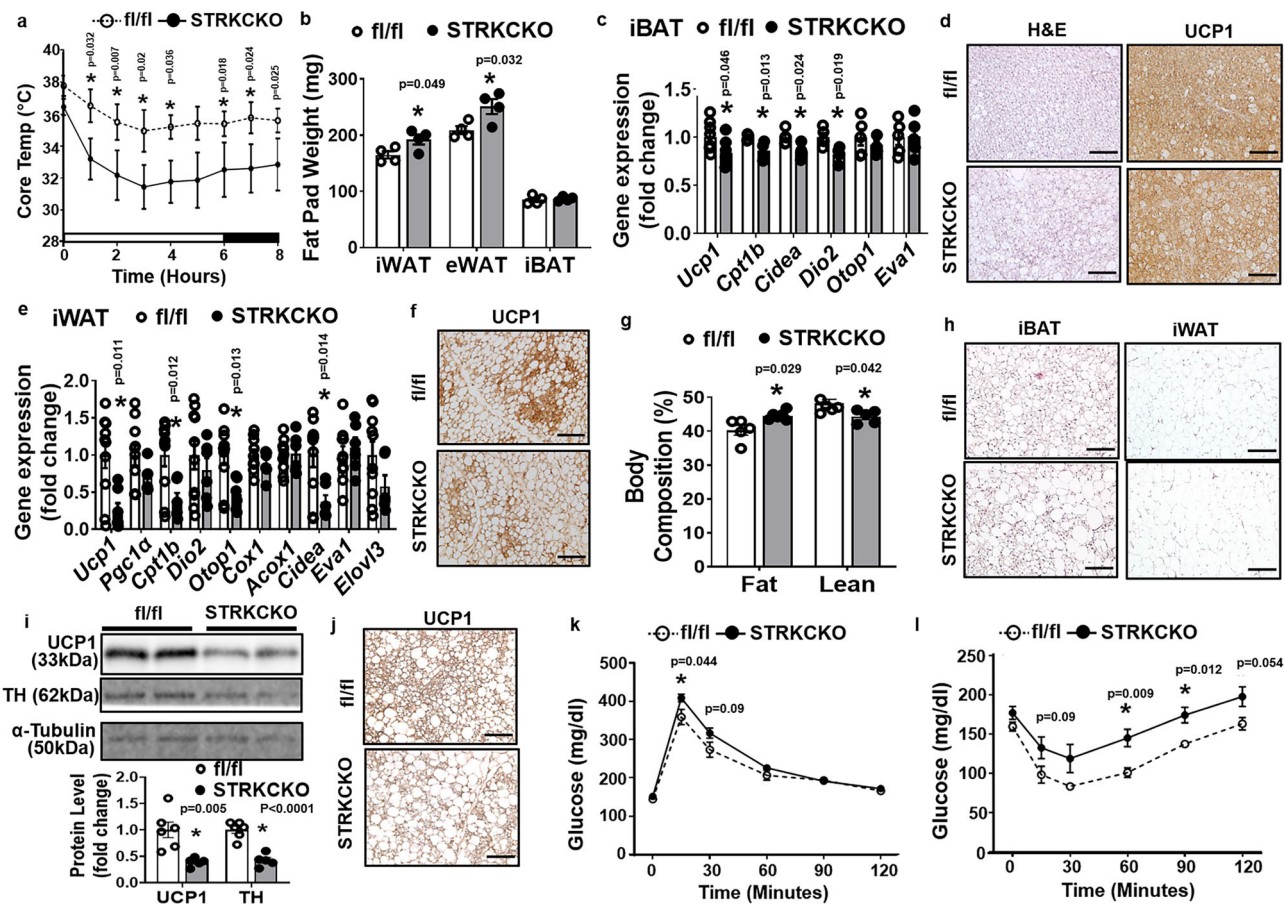

**Fig. 9 Mice with specific deletion of TRKC in sympathetic neurons exhibit impaired cold-induced thermogenesis and are prone to diet-induced obesity.**
**a–f** Core body temperature (**a**, n = 4/group, * indicates statistical significance between fl/fl and STRKCKO with unpaired two-tailed t test), fat pad mass (**b**, n = 4/group, * indicates statistical significance between fl/fl and STRKCKO with unpaired two-tailed t test), Ucp1 and other thermogenic gene expression in iBAT (**c**, for Ucp1, fl/fl n = 9, STRKCKO n = 7; for other genes, fl/fl n = 5, STRKCKO n = 4, * indicates statistical significance between fl/fl and STRKCKO with unpaired two-tailed t test), Representative H&E and UCP1 immunostaining in iBAT (**d**, from three replicate animals/group, scale bar = 75 μm), Ucp1 and other thermogenic gene expression in iWAT (**e**, fl/fl n = 10, STRKCKO n = 6, * indicates statistical significance between fl/fl and STRKCKO with unpaired two-tailed t test), and representative UCP1 immunostaining images showing UCP1-positive beige adipocytes in iWAT (**f**, from three replicate animals/group, scale bar=150 μm) of fl/fl and STRKCKO mice during a 7-day cold challenge. **g–l** Body composition (**g**, n = 5/group, *indicates statistical significance between fl/fl and STRKCKO with unpaired two-tailed t test), representative H&E staining of iBAT and iWAT (**h**, from three replicate animals/group, scale bar for iBAT = 75 μm, and for iWAT = 150 μm), UCP1 and TH protein levels in iBAT (**i**, fl/fl n = 6, STRKCKO n = 5, * indicates statistical significance between fl/fl and STRKCKO with unpaired two-tailed t test), representative UCP1 immunostaining in iBAT (**j**, from three replicate animals/group, scale bar = 75 μm), GTT (**k**, fl/fl n = 5, STRKCKO n = 6, * indicates statistical significance between fl/fl and STRKCKO with unpaired two-tailed t test) and ITT (**l**, fl/fl n = 5, STRKCKO n = 6, * indicates statistical significance between fl/fl and STRKCKO with unpaired two-tailed t test) of fl/fl and STRKCKO mice fed a HFD. All data are expressed as mean ± SEM.

Neurotrophins are then retrogradely transported from the target along the axons into the neuronal cell body, where they regulate neuron function. This mode of action continues during development and throughout adult life in order to maintain the normal differentiation, growth, and function of the neuron and thus maintaining sufficient PNS innervation in target tissues[29,30,53].

The role of BDNF in the regulation of energy balance has been extensively studied, which was largely contributable to its regulation of food intake through CNS mechanisms[54–57]. It was reported that resistance training and combined high intensity exercising and resistance training increased plasma NT-3 levels in overweight subjects[58]; and administration of NT-3 in db/db mice reduced their blood glucose level[59]. In addition, circulating NT-3 level was reported to be negatively correlated with total cholesterol and low-density lipoprotein (LDL)–cholesterol levels[60], indicating a role of NT-3 in metabolic regulation. However,

whether and how NT-3 regulates energy balance has been largely unknown.

A recent study demonstrated that S100, a BAT-derived secretory protein, promotes SNS innervation into adipose tissue. However, it is not clear whether S100 exerts the neurotrophic effect on SNS by itself or needs to work with other neurotrophic factors[21]. Interestingly, a newly identified hepatokine Tsukushi (TSK) has been shown to suppress brown fat thermogenesis via downregulating sympathetic innervation[61]. However, the exact mechanism underlying the inhibitory effect of TSK on SNS innervation is not entirely clear. Our study demonstrates that NT-3 is a previously unidentified fat-derived neurotrophic factor that promotes SNS innervation through its direct effect on neurite growth, resulting in enhanced brown/beige fat thermogenesis and energy expenditure.

Existing literature supports an important role of target-derived NT-3 in the regulation of normal SNS neuron function, growth, and target tissue innervation. Homozygous NT-3-deficient mice

have significantly depleted peripheral sympathetic neurons in SG[24], and reduced norepinephrine levels in the heart[25]. In addition, in homozygous NT-3 knockout mice, sympathetic fibers fail to invade the pineal gland and the external ear postnatally; whereas in NT-3 heterozygous mice, sympathetic fibers invade the pineal gland, but fail to branch properly and form a ground plexus[62]. On the other hand, cutaneous overexpression of NT-3 increases sympathetic neuron number in the SG and enhances hair follicle innervation[63]. We demonstrate here that BAT and WAT also produce NT-3 that regulates SNS innervation and activation in BAT and WAT in response to the developmental and environmental cues.

In this study, we have used pharmacological NT-3 injection and genetic models to delineate the physiological function of NT-3. The combination of our AdNT3-Tg, AdNT3-KI, and NT-3 injection models help provide an overall picture of the physiological role of the fat-derived NT-3 in the regulation of energy metabolism. Characterization of our transgenic AdNT3-Tg mice that have overexpression of *NT-3* in fat tissue without a systemic increase of NT-3 protein in the circulation demonstrates that increasing NT-3 locally in fat tissue should have a sufficient and direct effect on SNS innervation, increasing beige cell formation and thereby promoting energy expenditure. In addition, although AdNT3-KI mice have increased NT-3 in circulation, the obesity resistance observed in AdNT3-KI mice appears to be primarily due to increased energy expenditure stemming from enhanced sympathetic innervation and thermogenesis in adipose tissue. HFD-fed AdNT3-KI mice do not display any differences in food intake but have increased energy expenditure. Studying gene expression profiles in the hypothalamus, liver, and muscle also has ruled out the contribution of NT-3/TRKC signaling in these tissues in the regulation of energy homeostasis. In addition, there is also no difference in NE content and NETO in muscle, kidney, and heart between TRKC + /− and WT controls after a 16-h cold challenge. Thus, our data suggest that brown and white adipose tissues may be primary targets of NT-3/TRKC signaling in the regulation of energy homeostasis.

It appears that NT-3/TRKC's signal may directly stimulate sympathetic neuron axonal growth, which may contribute to its stimulatory effect on SNS innervation into fat tissue. Through RNA-seq analysis, we have identified genes in several important pathways involved in neuronal axonal growth are regulated in an opposite direction in SG of AdNT3-Tg and TRKC + /− mice, including the coordination of actin filaments and micro-tubules, the dynamic cytoskeletal polymers, and "building blocks" that promote shape change and locomotion during the peripheral nerve axonal outgrowth and elongation; family of proteins with GTPase activities that regulate actin motility, polymerization, and stabilization during axonal initiation and elongation; regulation of glucose metabolism that provides the energy needed for neuronal activity and axonal growth; protein sorting and transporting machineries; and proteins involved in axonal transport and translation of mRNAs[38,39]. In addition, our data also point to a potential link between inflammation and neuronal axonal growth. These genes and pathways may represent important NT-3/TRKC downstream targets that warrant future investigation.

In summary, we have discovered that the fat-derived neurotrophic factor NT-3 and its receptor TRKC are key regulators of SNS growth and innervation in adipose tissue (Fig. 10). Our data indicate that NT-3 expression might be responsible for the induction of developmentally induced beige adipocytes via sympathetic innervation in adipose tissue. We also demonstrate that NT-3 promotes cold-induced thermogenesis, enhances systemic energy expenditure, and prevents diet-induced obesity.

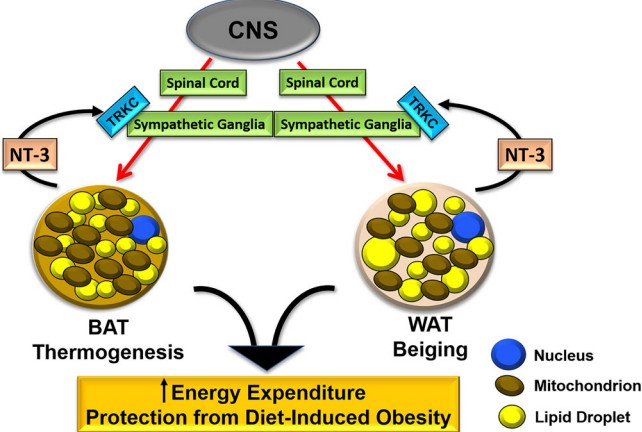

**Fig. 10 Schematic illustration of the role of fat-derived NT-3 and its receptor TRKC at sympathetic ganglia in the regulation of SNS innervation in adipose tissue.** Briefly, the neurotrophic factor NT-3 secreted from adipose tissue acts through its receptor TRKC in sympathetic ganglia to promote SNS innervation in brown and white adipose tissues, which in turn stimulates BAT thermogenesis and WAT beiging, leading to increased energy expenditure and protection from diet-induced obesity.

## Methods

**Mice.** Mice with haploinsufficiency of TRKC (TRKC + /−) were purchased from the Jackson Laboratory (#002481, Bar Harbor, ME). Mice with sympathetic-specific deletion of TRKC (STRKCKO) were generated by crossing the TRKC-floxed mouse model (fl/fl) where the TRKC kinase ATP-binding domain is flanked with loxP sites[35] (Jackson Laboratory #022364) with TH-Cre mice where Cre-recombinase expression is under the control of the sympathetic marker tyrosine hydroxylase (TH) promoter[36](Jackson Laboratory #008601) to achieve specific deletion of TRKC in sympathetic neurons.

Reporter mice that express enhanced yellow fluorescent protein (EYFP) under the control of TRKC promoter (TRKC-Cre::R26-stop-EYFP) were generated by crossing TRKC-Cre mice[64] (NIH-supported Mutant Mouse Resource and Research Centers (MMRRC), #000364-UCD) with mice expressing a loxP-flanked STOP sequence followed by EYFP gene inserted into the Gt(ROSA)26Sor locus[65](Jackson Laboratory #006148).

To generate mice with NT-3 overexpression in adipocytes (AdNT3-Tg), a bacterial artificial chromosome (BAC) containing the mouse adiponectin gene was used to create the adiponectin promoter-NT-3 overexpressing construct. The full-length coding sequence of the mouse NT-3 gene with a SV40 polyA signal sequence was PCR-amplified and inserted into the ATG position on exon 2 of the adiponectin gene in BAC using homologous recombination. The DNA fragment containing 5-kb adiponectin promoter and NT-3 coding sequence was subcloned into pCR-Blunt, excised, and microinjected into pronuclei of fertilized embryos of C57BL/6J mice at our Georgia State University Transgenic and Gene Targeting Core (Atlanta, GA). Transgenic founders were identified by PCR amplification of a ~900 bp fragment from tail DNA.

To alternatively achieve specific overexpression of *NT-3* in adipose tissue, we used a gene-targeting strategy to insert the *NT-3* transgene into the *Rosa26* locus, which is the most used locus for the site-specific integration of transgenes due to high efficiency of homologous recombination[27]. The murine *NT-3* cDNA was cloned into a *Rosa26* locus targeting vector[27] purchased from Addgene (Cambridge, MA). Key features of the complete targeting construct as depicted in Supplementary Fig. 8a include the following: (1) a strong CAG promoter allowing ubiquitous expression of cDNA construct placed behind it; (2) a loxP-flanked transcriptional blocker (Stop) that contains the transcription-blocking sequence; (3) the coding sequence for *NT-3* with woodchuck hepatitis virus posttranslational regulatory element (WPRE); and followed by (4) a flippase (*Flp*) recognition target (FRT)-flanked neomycin selecting cassette (Neo). With this design, the strong CAG promoter can only drive *NT-3* expression when the transcriptional block is removed by a tissue-specific Cre, therefore achieving tissue-specific overexpression. The DNA construct was electroporated into ES cells for homologous recombination targeting the endogenous *Rosa26* locus. About 30% of ES cell colonies were identified as positive by Southern blotting showing a wild-type band at 4.3 kb and a targeted band at 5.5 kb (Supplementary Fig. 8b). Correctly targeted ES cells were injected into blastocysts by our Transgenic and Gene Targeting Core (Georgia State University, Atlanta, GA). The resulting chimeric founders were crossed with C57BL/6J mice to achieve germline transmission of the knock-in allele, which were further bred with Flp mice[66](Jackson Laboratory, Stock No. 009086) to delete the neomycin cassette. The resulting Rosa-NT-3^fl/+ mice were crossed with Adiponectin-Cre mice to generate mice with adipocyte-specific NT-3

knock-in mice (Adiponectin-Cre::Rosa-NT-3$^{fl/+}$, or AdNT3-KI) by deleting the loxP-flanked Stop sequence.

For NT-3 injection experiment, postnatal C57BL/6J mice at age of day 20 were intraperitoneally (i.p.) injected with recombinant human NT-3 (50 μg/kg body weight, R&D Systems, Minneapolis, MN) daily for 10 days, while 6 weeks old C57BL/6J mice on HFD were i.p. injected with the same dose of recombinant human NT-3 (OriGene Technologies, Rockville, MD) every day for the first two weeks, then three times per week for 8 weeks. Similar NT-3 doses and frequencies have been reported in the literature in preventing experimental pneumococcal meningitis-induced neuron loss[67] and in modulating nociceptive threshold and the release of substance P from rat spinal cord[68].

For Fast Blue injection, TRKC-Cre::R26-stop-EYFP, AdNT3-Tg, TRKC+/− mice and their littermate controls were anesthetized via isoflurane, and a dorsal or ventral 2 cm incision was made to expose IBAT or iWAT. The retrograde tracer Fast Blue (FB) (2%; Polysciences, PA) was injected with a microsyringe into 5–10 separate loci (1 μl/locus) of each adipose tissue. Animals were given 10–14 days to recover and allow for retrograde Fast Blue transport.

All animal procedures were approved by the Institutional Animal Care and Use Committee of Georgia State University and were in compliance with the Public Health Service and the United States Department of Agriculture guidelines.

**Metabolic measurement.** All mice were housed with a 12/12 h light–dark cycle in temperature- and humidity-controlled rooms with free access to water and food (ambient temperature: 20–22 °C, thermoneutral temperature: 30 °C; humidity: 30–70%). TRKC+/−, STRKCKO, AdNT3-Tg, AdNT3-KI mice and their respective littermate controls were fed either low fat (LF) (Research Diets D12450B, 10% calorie from fat) or high-fat (HF) (Research Diets D12492, 60% calorie from fat) diet for up to 30 weeks. In most of the experiments, mice were housed at ambient temperature (20–22 °C). Some experiments were conducted under thermoneutrality (30 °C) to avoid any nonshivering thermogenesis that may be induced by mild cold stress in mice housed under ambient room temperature (20–22 °C) as we recently reported[49], which may serve as a confounding factor in assessing diet-induced thermogenesis[4,50]. Various metabolic phenotypes were characterized as follows. (1) Body weight was monitored weekly. (2) Food intake, energy expenditure, and activity levels were measured using PhenoMaster metabolic cage systems (TSE Systems, Chesterfield, MO). (3) Body composition was analyzed using a Minispec NMR body composition analyzer (Bruker BioSpin Corporation; Billerica, MA). (4) Insulin sensitivity was determined by glucose tolerance and insulin tolerance tests (GTT and ITT, respectively) as we previously described[69]. (5) Pyruvate tolerance test was performed as described[70]. Briefly, animals fasted for 16 h, and pyruvate was injected intraperitoneally at 1 g/kg body weight. Blood glucose was measured OneTouch Ultra Glucose meter (LifeScan, Milpitas, CA) before and 15, 30, 60, 90, and 120 min after pyruvate injection. At the end of dietary treatment, BAT and WAT tissues were collected for further analysis of brown fat/beige adipocyte thermogenic program including gene expression, protein expression, and immunohistochemistry.

**Cold exposure.** TRKC-Cre::R26-stop-EYFP, AdNT3-Tg, TRKC+/−, STRKCKO mice and their respective littermate controls were subjected to a chronic 7-day cold challenge (5 °C). At the end of the experiment, tissues were collected for further analysis of brown fat/beige adipocyte thermogenic program, including gene expression, protein expression, and immunohistochemistry. In some experiments, core temperature was measured by a temperature transponder implanted into the peritoneal cavity as we previously described[37]. Briefly, 2 weeks prior to the start of cold exposure, animals were anesthetized and a small incision was made along the ventral midline to expose the peritoneal cavity. Sterile temperature transponders (IPTT-300, BioMedic Data Systems, Seaford, DE) were implanted into the peritoneal cavity, and the incision was closed with sterile dissolvable sutures and sterile wound clips. After recovery, animals were subjected to a chronic 7-day cold challenge (5 °C). Core body temperature was measured every 1–2 h on the designated days during the cold exposure.

**Quantitative RT-PCR.** Total RNA from adipose tissues and SG were isolated using the Tri Reagent kit (Molecular Research Center, Cincinnati, OH)[69]. The expression of genes of interest was measured by a one-step quantitative RT-PCR with TaqMan Universal PCR Master Mix reagents (ThermoFisher Scientific, Waltham, MA) using an Applied Biosystems QuantStudio 3 real-time PCR system (ThermoFisher Scientific) as we previously described[69]. The mRNA quantitation was further normalized by the housekeeping gene cyclophilin. The sequences of the primer and probe pairs for UCP1 and cyclophilin are as follows. UCP1: forward 5'-CACCTTCCCGCTG-GACACT-3'; reverse 5'-CCCTAGGACACCTTTATACCTAATGG-3'; probe 5'-AGCCTGGCCTTCACCTTGGATCTGA-3'. Cyclophilin: forward 5'-GGTGGA-GAGCACCAAGACAGA-3'; reverse 5'-GCCGGAGTCGACAATGATG-3'; probe 5'-ATCCTTCAGTGGCTTGTCCCGGCT-3'. The TaqMan primers/probes for all the other genes were purchased from Applied Biosystems (ThermoFisher Scientific).

**RNA-seq analysis.** RNA-seq library preparation, sequencing, and basic bioinformatics data analysis from SG were performed by BGI Americas (Cambridge, MA), which specializes in high-throughput RNA/transcriptome sequencing and bioinformatics data analysis[71,72]. Briefly, after total RNA extraction and digestion with DNase I, mRNA were enriched with the oligo(dT) magnetic beads, fragmented (about 200 bp), and used for cDNA synthesis with random hexamer–primer. The double-stranded cDNA was ligated with sequencing adaptors and PCR-amplified. RNA-seq libraries were sequenced using Illumina HiSeq$^{TM}$ 2000 (SE50). For quality control, RNA and library preparation integrity were verified using Agilent 2100 BioAnalyzer system and ABI StepOnePlus Real-Time PCR System.

For bioinformatics analysis, raw reads were filtered to remove adaptor sequences and low-quality data using SOAPnuke (v1.5.2, https://github.com/BGI-flexlab/SOAPnuke)[73] and mapped to reference sequences (University of California Santa Cruz Mouse Genome Browser mm9 Assembly) using Hierarchical Indexing for Spliced Alignment of Transcripts (HISAT2, v2.0.4, http://www.ccb.jhu.edu/software/hisat/index.shtml)[74,75]. Reads per kilobase per million reads (RPKM) were calculated to represent the gene expression level, and were used for comparing differentially expressed genes (DEGs) among groups. To identify genes regulated in the opposite direction in SG from thoracic T1 to T4 levels of AdNT3-Tg and TRKC+/− mice after 1-day cold challenge, the RNA-seq datasets were merged based on Entrez gene ID. Genes up- or downregulated in opposite direction in AdNT3-Tg and TRKC+/− mice were identified using a cutoff value of absolute log$_2$ fold change (log2FC) ≥ 0.5.

**Immunoblotting and NT-3 level measurement.** Protein expression in adipose tissue was measured by immunoblotting as we described[69,76,77]. Fat tissues were homogenized in a modified radioimmunoprecipitation assay (RIPA) lysis buffer supplemented with 1% protease inhibitor mixture and 1% phosphatase inhibitor mixture (Sigma-Aldrich, St. Louis, MO). Tissue lysates were resolved by SDS-PAGE. Proteins on the gels were transferred to nitrocellulose membranes (Bio-Rad, Hercules, CA), which were then blocked, washed, and incubated with various primary antibodies, followed by Alexa Fluor 680-conjugated secondary antibodies (Life Science Technologies). The blots were developed with a Li-COR Imager System and analyzed with Li-COR Image Studio Software (version 2.1, Li-COR Biosciences, Lincoln, NE). The following primary antibodies were used: UCP1 (1:1000, Abcam, ab23841), TH (1:1000; AB152, EMD Millipore, Temecula, CA), NT-3 (1:1000, AF-267-NA, R&D Systems), pHSL (1:1000, 4126S, Cell Signaling Technology), HSL (1:1000, 4107S, Cell Signaling Technology), and α-tubulin (1:1000, Advanced BioChemicals, ABCENT4777). The antibody information is listed in Supplementary Table 1.

Tissue and serum NT-3 content was measured by ELISA (ABCE-EL-M2438, Advanced BioChemicals, Lawrenceville, GA).

**Immunohistochemistry (IHC).** Fat tissue was fixed in 10% neutral formalin, embedded in paraffin, and cut into 5-μm sections, which were either processed for hematoxylin and eosin (H&E) staining or immunostaining with a UCP1 antibody (1:500, Abcam, ab10983) as we previously described (Supplementary Table 1)[16,37]. For immunostaining, the secondary antibody was conjugated with horseradish peroxidase (HRP) from Vector Labs (Vectastain ABC HRP kit, Vector Labs, PK-6100) and was developed with the 3,3'-diaminobenzidine (DAB) HRP substrate kit (Vector Labs, SK-4100).

SG were sectioned and immunostained as we described previously[37]. Briefly, SG were carefully harvested and transferred to an 18% sucrose solution in 0.1 M PBS containing 0.1% sodium azide at 4 °C. All ganglia were then sectioned longitudinally at 10-μm-thick sections using a cryostat. They were directly mounted onto slides (Superfrost Plus; VWR International, West Chester, PA) in four series with every fifth section on the same slide. Sections were then incubated with rabbit anti-Tyrosin hydroxylase (TH, 1:500, AB152, Millipore, MA) or rabbit anti-cFos (1:500, ABE457; Millipore, MA) antibodies overnight. In some experiments, sections were also incubated with chicken anti-GFP antibodies (1:500, GFP1010, Aves labs). Sections were then incubated with Cy3-donkey anti-rabbit (1:500; Jackson Immunoresearch, West Grove, PA) or Alexa Fluor 488-Donkey Anti-Chicken (1:500; Jackson Immunoresearch, West Grove, PA) secondary antibodies for 2 h (Supplementary Table 1). Sections were mounted onto slides and coverslipped using ProLong Gold Antifade Reagent (Life Technologies, Grand Island, NY). Images were captured using an Olympus DP73 photomicroscope and CellSens software (version 1.6) (Olympus, Waltham, MA). The captured images were evaluated with the aid of Image J software (version 1.5.2). After two images were overlaid, exhaustive counts of FB- and TH (or cFos)-single neurons as well as FB with TH (or cFos)-colocalized neurons were performed by use of the manual tag feature of Image J software (version 1.5.2) in every fifth section of the ganglia to eliminate the likelihood of counting the same neuron twice. The neurons were considered positively labeled based on the fluorescent intensity, cell size, and shape. Percentage of positively single- and double-labeled neurons in the ganglia were averaged across each examined region from all mice.

**Assessment of sympathetic nerve innervation with adipose tissue whole-mount clearing and immunostaining.** A whole-mount adipose tissue clearing with the Adipo-Clear approach[22] was conducted to allow immunostaining, followed by three-dimensional visualization and quantitation of sympathetic nerve innervation by inverted confocal fluorescence microscopy. Briefly, adipose tissue was processed with a series of steps including dehydration, delipidation, and

permeabilization, which was further stained with primary anti-TH antibodies (AB152, 1:1000, Millipore) and secondary antibodies (Cy™3 AffiniPure Donkey Anti-Rabbit IgG (H + L), 1:2000, Jackson ImmunoResearch, 711-165-152) (Supplementary Table 1). Tissues were further cleared with dibenzyl ether (DBE) and imaged with Zeiss 710 NLO Laser Scanning Confocal Microscope (Optical Microscopy Core, Georgia Institute of Technology, Atlanta, GA). Three-dimensional image reconstruction, sympathetic nerve fiber tracing, and nerve density quantification, including nerve branching/intersection points and average neurite length, were analyzed using the Imaris Image Analysis Software (version 9.5.1), and the volume of each WAT was obtained using the Surface tool within the Imaris Image Analysis Software to correct the size difference of each sample. (Oxford Instruments, imaris.oxinst.com).

**Sympathetic neuronal culture and neurite growth measurement**. SG at thoracic T1–T4 levels from 6- to 8-week-old mice were dissected, digested with collagenase I followed by Trypsin digestion. Dispersed neurons were cultured in complete neurobasal media (Fisher 10888022) with 1X B-27 (Fisher 17504044) and 200 μM L-glutamine (Fisher 25030149) in poly-D-lysine and Laminin coated dishes. Cells were cultured for 48 h with phosphate-buffered saline (PBS), NT-3 (100 ng/ml, R&D Systems, Minneapolis, MN) or CXCL10 (100 ng/ml, R&D Systems) treatment, re-plated, and further cultured for an additional 3–6 h with PBS, NT-3 (100 ng/ml) or CXCL10 (100 ng/ml) treatment. A similar NT-3 dose has been used in neuronal cultures previously[78]. Cells were then immunostained with Alexa Fluor® 488-conjugated antibodies against neurite growth marker βIII-tubulin (1:400, AB15708A4, Millipore) (Supplementary Table 1)[79]. Neurite growth was then quantitated by Image J (version 1.5.2) with Neuron J plugin (version 1.4.3)[80]. Between 40 and 168 neurons were analyzed in each group, and neurite number and neurite length were normalized by the number of neurons in each group when doing comparisons between groups.

**Norepinephrine turnover measurement**. Basal norepinephrine (NE) level and NE turnover rate (NETO) in adipose tissues were measured as we previously described[37]. Briefly, mice were handled daily for 1 week prior to the start of the NETO experiment to adapt them to handling and reduce stress-induced NE release. On the day that NETO was measured, mice were subjected to cold exposure at 5 °C for 16 h and NETO was measured during the last 4 h of the experiment. mice were injected intraperitoneally with α-methyl-p-tyrosine (250 mg/kg α-MPT; Sigma-Aldrich), an active competitive inhibitor for TH, which is the rate-limiting enzyme for NE production, thus preventing the synthesis of catecholamine. A supplemental dose of α-MPT (125 mg/kg) was given 2 h after the initial dose to ensure the inhibition of catecholamine synthesis. Two hours after the second α-MPT injection, mice were euthanized and various tissues were quickly harvested, weighed, frozen in liquid nitrogen, and stored at −80 °C until NE extraction.

To obtain baseline NE values, one-half of mice from each group were euthanized without receiving α-MPT injections 4 h before the conclusion of the study. The adipose tissues were processed and extracted for NE with dihydroxybenzylamine (Sigma-Aldrich) as an internal control for extraction efficiency. NE content and NETO were measured as described previously[81]. NETO was calculated using the following formula: $k = (\lg[NE]_0 - \lg[NE]_4)/(0.434 \times 4)$ and $K = k[NE]_0$, where $k$ is the constant rate of NE efflux, $[NE]_0$ is the initial NE concentration, $[NE]_4$ is the final NE concentration, and $K = $ NETO.

**Statistical analysis**. All data are expressed as mean ± SE. All graphs were made with GraphPad Prism (v9.1.2, GraphPad Software, San Diego, CA). Differences between groups were analyzed for statistical significance by unpaired two-tailed Student's $t$ test, one-way or two-way ANOVA followed by Turkey's multiple comparisons test as described under the figure legends for each figure using SPSS Statistical Software (v27.0.1.0, IBM, Armonk, NY). For all experiments, differences among groups were considered statistically significant at $P < 0.05$.

**Reporting summary**. Further information on research design is available in the Nature Research Reporting Summary linked to this article.

## Data availability
The RNA-seq data have been deposited to Gene Expression Omnibus (GEO) database with the accession code GSE173503. All data generated in this study are included in this article. Source data are provided with this paper.

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

## Acknowledgements

This work is supported by NIH grants R01DK107544, R01DK118106, and R01DK125081, and American Diabetes Association (ADA) grant 1-18-IBS-260 to B.X.; NIH grants R01DK115740 and R01DK118106, and ADA grant 1-18-IBS-348 to H.S.; NIH grant R01DK116496 and ADA grant 1-18-IBS-346 to L.Y.

## Author contributions

X.C. performed experiments of AdNT3-Tg mouse generation and characterization, TRKC +/− mice characterization, NT-3 injection in adult mice fed a high-fat diet, all sympathetic innervation with Adipo-clear and imaging, sympathetic neuron culture and neurite growth, generation and characterization of STRKCKO mice, and performed the data analysis of the experiments; J.J. performed experiments of AdNT3-KI mouse generation and characterization, and performed the data analysis of the experiments; RW performed experiments with NT-3 injection of postnatal P20 mice and characterization of TRKC +/− mice, and performed the data analysis of the experiments; Q.C. performed sympathetic neuron culture experiments and data analysis (along with X.C.); Q.C., F.L., K.L., and S.W. assisted in various experiments; H.D.S. performed bioinformatics analysis of the RNA-seq data; G.S. and L.Y. contributed to study design, technical inputs and review/edits on manuscript; H.S. and B.X. conceived and designed the study and wrote the manuscript.

## Competing interests

The authors declare no competing interests.
