## [Peer Review File · Nature Communications]

Reviewers' Comments:

Reviewer #1:

Remarks to the Author:

The current study discovered for the first time that fat-derived neurotrophic factor NTF3 and its receptor TRKC regulated SNS growth and innervation in adipose tissue. It's of great interest to us that this study demonstrated a role of SNS-targeted tissue in regulating the development and activation of SNS. However, there are still some questions:

1. In figure 1C, I can not see significant downregulation of mean nerve fiber density, length or branching points in WAT of 3M mice compared to P20 pups.
2. In figure 2J, I think the up-regulation of UCP1 in the NTF3 treatment group is not significant.
3. How do you explain the fact that NTF3 was not increased in the iBAT and serum of AdNTF3-Tg mice?
4. All of the H&E, immunostaining figures should be labeled with scale bars or described in the figure legends.
5. In figure 5F, you should better add the the results of control groups.
6. I see that some parameters in the iBAT is not very significant. For example, there was no change in UCP1 protein levels and UCP1 immunostaining in iBAT of TRKC \pm mice. I wonder if iBAT is not important in the regulation of SNS growth or thermogenesis.
7. There are some spelling mistake in the Discussion part, such as "On the other hand", not "One the other hand" or "NTF3", not "NT3".

Reviewer #2:

Remarks to the Author:

Xin Cui and coworkers describe in this manuscript that the neurotrophic factor neurotrophin 3 (NTF3) and its receptor tropomyosin receptor kinase C (TRKC) are involved in the regulation of sympathetic nervous system growth and innervation of adipose tissue. They found that NTF3 is highly expressed in brown adipocytes. In transgenic mice with adipocyte-specific NTF3 overexpression, adipose tissue sympathetic innervation is increased whereas mice with TRKC haploinsufficiency (TRKC \pm) have reduces sympathetic innervation in adipose tissue. Using pharmacological or genetic approaches, the authors found that NTF-3 promotes beige adipocyte development, enhances cold-induced thermogenesis and protects against diet-induced obesity. The opposite phenotype was observed in TRKC \pm mice. The studies are in principle well performed, data are novel and timely and add significantly to our understanding how adipose tissue innervation is regulated. There are a few comments that maybe considered:

Comments:

- 1) It is confusing that the authors state in different parts of the manuscript that they discovered neurotrophin-3 (NTF3), but there are several studies on neurotrophin-3 (NT-3) in the literature (summarized in *Handb Exp Pharmacol.* 2014;220:497-512). Is the previously reported neurotrophin-3 different from the one described in this manuscript?
- 2) Do mice with adipocyte-specific NTF3 overexpression exhibit a CNS phenotype?
- 3) It remains unclear how the authors characterize beige adipocytes and differentiate those from brown adipocytes.
- 4) What is meant by "...maintain the vigorous innervation of sympathetic nerves..." Is sympathetic innervation of adipose tissue really diminished in adult mice? Sympathetic nerve fibers should reach adipose tissue in close proximity of vessels. Therefore it remains unclear, how the proposed "sympathetic denervation" may physiologically occur.
- 5) It would be interesting to add a physiological read out to the reduced SNS innervation observed with imaging techniques in iWAT of 3-month-old mice. Was there any effect on lipolysis?
- 6) What would be the mechanism for increased lean body mass upon NTF3 pharmacological treatment?
- 7) Was body temperature measured in response to NTF-3 injections?
- 8) The proposed NTF3/TRKC signaling-related link between inflammation and neuronal axonal growth should be substantiated by data.

Minor comments:

- 1) The authors use both "neurotrophic" and "neurotropic" factor to describe NTF3. Is that on purpose?
- 2) Statements regarding the impact of the findings on obesity treatment should be toned down to a realistic level.
- 3) The sentence: "Although brown fat thermogenesis was traditionally viewed as a defense mechanism against cold in rodents, recent discovery of metabolically active brown fat in adult humans has implicated BAT thermogenesis as a promising therapeutic target for the treatment of obesity" is unclear - why should that be a contradiction?

Reviewer #3:

Remarks to the Author:

An interesting paper supported by a large body of work utilizing a good range of techniques and transgenic mouse models. The authors provide persuasive data revealing a novel NT-3/trkC signaling axis regulating sympathetic innervation of adipose tissue (white and brown fat). Using a variety of approaches they show that cold-induced thermogenesis in adipocyte tissue is augmented by NT-3/trkC signaling and is associated with enhanced sympathetic innervation. They propose that NT-3/trkC signaling, elevated sympathetic innervation and elevated cold-induced thermogenesis by adipose tissue can alter the body's energy metabolism to increase insulin sensitivity and reduce obesity. The work is clearly presented with a significant body of useful supplementary information. I have the following concerns:

1. The HFD studies would have been easier to interpret if a lean control had been included. Consequently, we don't really know the level of obesity being induced in these studies.
2. In all the studies the fed glucose level in the blood was presented. It would have been more informative to measure fasted blood glucose. In addition, more repetitive measures during the course of the study would have been more informative. Especially in the AdNT3-Tg mouse study.
3. The study with the AdNT3-Tg mouse line is probably the most informative aspect of the work. This mouse line used the adiponectin driver for specific expression in adipose tissue. However, how specific is this promoter? The literature reveals a number of studies revealing expression of adiponectin mRNA in skeletal muscle, myocardium and liver. Thus, there is the possibility of over-expression of NT-3 in these tissues. The mouse line should have been fully validated with data on NT-3 expression in these tissues compared between WT and AdNT3-Tg. While systemic expression of NT-3 in serum was not altered there remains the possibility of enhanced NT-3/trkC signaling in these organs/tissues that makes interpretation of the general effects on insulin sensitivity and body weight problematic.
4. The authors tried to complement the AdNT3-Tg study with an alternative mouse line, the AdNT3-KI mouse. This is laudable, however, the issue here is the very high systemic levels of NT-3 in these mice (Supp Fig. 7 E). This finding makes interpretation of the insulin sensitivity and body weight data very difficult (e.g. ability to link the findings to changes in cold-induced thermogenesis).
5. Interpretation of findings from the trkC +/- or STRKCKO mice reveals the same drawbacks as above. Here systemic effects on trkC/NT-3 signaling could contribute to energy metabolism and insulin sensitivity at multiple sites and thus the link with proposed effects of trkC signaling and adipose tissue and cold-induced thermogenesis is indistinct.
6. Which sympathetic ganglia were cultured in Figure 4 and were they adult neurons?
7. In Fig.4 for the cultures of WT vs trkC +/- were the neurons exposed to NT-3?
8. In Figure 5 it is not clear how the numbers of trkC positive neurons innervating iBAT from T2 ganglia is changing. Not discussed anywhere. Is the suggestion that trkC positive neurons not normally innervating the iBAT now innervate upon cold exposure? Or are normally trkC negative neurons becoming trkC positive in response to cold?
9. In Fig.5 (E) the % of total TH +ve or cFos +ve neurons innervating the iBAT are much higher than in the study depicted in parts A-D. This seems odd since they are actually trkC +/-.

In summary the authors provide a strong case supporting a novel NT-3/trkC signaling axis that

controls sympathetic innervation to mediate adipocyte phenotype and regulate the thermogenic response to cold. What is less clear is the impact of this process on whole body metabolism and insulin sensitivity. Sympathetic neurons, many expressing *trkC*, innervate multiple tissues/organs and so systemic alterations in NT-3 or local changes in NT-3/*trkC* signaling could impact insulin sensitivity and energy homeostasis. For example, an obvious possibility is altered NT-3 levels or *trkC* signaling in sympathetic fibers in the liver. The hypothalamic nuclei, in part, via the vagus nerve control liver function to regulate glucose production – it remains possible that some of the studies in this paper could be impacting this pathway. In the studies where systemic changes in NT-3 signaling were observed, e.g. Fig. 2 and Supp Fig. 4 or Supp Fig. 7+8, it is pertinent that these interventions caused the most robust effects on body weight and insulin sensitivity.

Minor comments:

The accepted abbreviation for neurotrophin-3 is NT-3 (not Ntf-3). Not wise to change a standard abbreviation first instigated by Yancopoulos et al in 1990.

Reviewer 1:

1. *In figure 1C, I can not see significant downregulation of mean nerve fiber density, length or branching points in WAT of 3M mice compared to P20 pups.*

Thank you for pointing this out. We have replaced **Figure 1C** with more representative images.

2. *In figure 2J, I think the up-regulation of UCP1 in the NTF3 treatment group is not significant.*

Thank you for pointing this out. Since UCP1 is such an abundant protein in iBAT, we have rerun the immunoblotting using less amount of lysates and updated **Figure 2J**.

3. *How do you explain the fact that NTF3 was not increased in the iBAT and serum of AdNTF3-Tg mice?*

We showed that *Nt-3* expression is much higher in iBAT than that of iWAT (**Fig 1D-F**). Possibly because of the high basal *Nt-3* expression in iBAT, *Nt-3* expression is not further increased in the AdNT3-Tg mice in iBAT.

To answer the second question, *Nt-3* expression is mildly increased in adipose tissue of AdNT3-Tg mice. Thus, the *Nt-3* overexpression in the AdNT3-Tg mice is probably restricted to local adipose tissue due to its mild overexpression in the adipose tissue without any changes in serum level.

We have incorporated these descriptions in the results part at page 9, first paragraph.

4. *All of the H&E, immunostaining figures should be labeled with scale bars or described in the figure legends.*

The scale bars have been added and also described in the figure legends.

5. *In figure 5F, you should better add the the results of control groups.*

The results of the control groups have been added in **Figure 7A** in the revised manuscript.

6. *I see that some parameters in the iBAT is not very significant. For example, there was no change in UCP1 protein levels and UCP1 immunostaining in iBAT of TRKC^{+/-} mice. I wonder if iBAT is not important in the regulation of SNS growth or thermogenesis.*

Thank you for pointing this out. We did observe that there was no change in UCP1 protein and immunostaining iBAT of TRKC^{+/-} mice compared to that of WT littermates after cold exposure (**Supplemental Figure 17**).

We think there are probably two reasons that can explain this. First, TRKC^{-/-} mice die during early postnatal period, thus, we used TRKC^{+/-} mice that retain one allele of

TRKC. Since there is high basal level of NT3/TRKC signaling in iBAT, it is possible that one allele of TRKC could still be sufficient to maintain proper NT3/TRKC signaling thus sympathetic innervation in iBAT. Second, it is possible that NT3/TRKC signaling may be more important for beige adipocyte function.

These descriptions have added to the results part at page 20, first paragraph.

7. *There are some spelling mistake in the Discussion part, such as "On the other hand", not "One the other hand" or "NTF3", not "NT3".*

Thank you for pointing this out. The spelling has been corrected. In addition, per the request of the second reviewer, and to be consistent with the nomenclature in the literature, all “NTF3” in the manuscript has been changed to “NT-3”.

Reviewer 2.

1. *It is confusing that the authors state in different parts of the manuscript that they discovered neurotrophin-3 (NTF3), but there are several studies on neurotrophin-3 (NT-3) in the literature (summarized in Handb Exp Pharmacol. 2014;220:497-512). Is the previously reported neurotrophin-3 different from the one described in this manuscript?*

Thank you for pointing this out. To clarify this question, NTF3 described in this manuscript is the same as NT-3 mentioned in the literature; NTF3 is the official symbol whereas NT-3 is one of the aliases commonly used in the literature. Thus, to keep consistent with the current literature, we have changed NTF3 to NT-3 throughout the manuscript.

2. *Do mice with adipocyte-specific NTF3 overexpression exhibit a CNS phenotype?*

Thank you for pointing this out. However, we Don't believe this is the case. First, mice with adipocyte-specific NT-3 overexpression (AdNT3-Tg and AdNT3-KI mice), the haploinsufficiency of TRKC (TRKC^{+/-}), and sympathetic specific TRKC deletion (STRKCKO) did not have a difference in food intake when compared to their wild type or flox littermate controls; this, in a way, argues against a CNS phenotype.

Second, we have now measured NT-3 expression in various tissues in the AdNT3-Tg and AdNT3-KI mice. We found NT-3 expression was not increased in the brain areas, including arcuate, paraventricular, ventromedial and lateral hypothalamus, cortex and hippocampus. These results were added in **Supplemental Figures 5F** (AdNT3-Tg mice), **8E** (AdNT3-KI), and in the results part at page 9, first paragraph, and page 12, first paragraph.

Third, we have also measured neuropeptide expressions in the hypothalamus of AdNT3-Tg and AdNT3-KI mice. We found there was no difference in the expression of neuropeptides pro-opiomelanocortin (*Pomc*), neuropeptide Y (*Npy*), and agouti related neuropeptide (*Agrp*) in arcuate hypothalamus (ARC) of AdNT3-Tg and WT littermates fed HFD (added in **Supplemental Figure 6D**) and between AdNT3-KI and WT littermates fed HFD (added in **Supplemental Figure 11A**). The description of these results has been added in the results part at page 10, second paragraph, and page 12, last paragraph to page 13, first paragraph.

3. *It remains unclear how the authors characterize beige adipocytes and differentiate those from brown adipocytes.*

Beige and brown adipocytes express many similarities, including abundant cristae-dense mitochondria that express UCP1 and multilocular lipid droplets. They can be differentiated based on certain criteria.

First, anatomically, traditional brown adipocytes are present in discrete regions in the body, including interscapular, axillary, and perirenal BAT depots in mice; whereas beige adipocytes are dispersed throughout white adipose tissue depots and can be induced by adrenergic stimulation.

Second, recent studies have identified some beige adipocyte markers that could potentially distinguish beige adipocytes from traditional brown adipocytes. However, evidence also suggests that more studies may be needed to truly identify those unique beige adipocyte markers.

Thus, in this study, we have distinguished beige adipocytes from brown adipocytes using their anatomic location, i.e., brown adipocytes are those located in iBAT that constantly express UCP1. Beige adipocytes are those located in inguinal WAT (iWAT), epididymal WAT (eWAT) or retroperitoneal WAT (rWAT), which are induced during cold exposure.

4. *What is meant by "...maintain the vigorous innervation of sympathetic nerves..." Is sympathetic innervation of adipose tissue really diminished in adult mice? Sympathetic nerve fibers should reach adipose tissue in close proximity of vessels. Therefore it remains unclear, how the proposed "sympathetic denervation" may physiologically occur.*

Recent studies from Dr. Jeffrey Friedman's group showed that HFD feeding in adult mice led to a decrease in sympathetic innervation in WAT and BAT (Wang et al., Nature . 2020 Jul;583(7818):839-844). Thus, the decline of sympathetic innervation in adipose tissue under physiological conditions is not uncommon, and can be initiated by HFD feeding, or, in our case, aging.

The neurotrophic factor NT-3 is important for maintaining neuron function and axonal growth, and our data suggest that NT-3 expression in adipose tissue declines in adult mice compared to mice at postnatal 20 days of age. Thus, our data suggest that the NT-

3/TRKC signaling may be important in regulating sympathetic innervation in adipose tissue. The decline of NT-3 expression in adipose tissue and NT3/TRKC signaling in sympathetic neurons that innervate adipose tissue may contribute to reduced sympathetic innervation in adipose tissue.

These descriptions have added to the Discussion part at page 23, last paragraph.

5. *It would be interesting to add a physiological read out to the reduced SNS innervation observed with imaging techniques in iWAT of 3-month-old mice. Was there any effect on lipolysis?*

Thank you for pointing this out. To answer this question, we have tested adipose tissue lipolysis capacity in 20-day and 3-month old mice in response to an acute cold exposure at ~10°C for 6 hours. These data were incorporated into **Supplemental Figure 1D and 1E**. We found whereas there was no difference in serum non-esterified fatty acids (NEFA) level in 20-day and 3-month old mice housed at room temperature, serum NEFA level was reduced in 3-month old mice compared to that of 20-day old mice after a 6-hour cold exposure (Supplemental Fig 1D). This corresponds to slightly reduced hormone sensitive lipase (HSL) phosphorylation (pHSL) in iBAT and more profoundly reduced pHSL in iWAT of 3-month old mice after the cold exposure as compared to that of 20-day old mice (**Supplemental Fig 1E**). Thus, our data suggest that reduced sympathetic innervation in the fat tissues in older mice results in reduced lipolysis capacity in response to cold exposure.

We have also incorporated these results at page 6, second paragraph.

6. *What would be the mechanism for increased lean body mass upon NTF3 pharmacological treatment?*

For Fig 2E, we have originally presented body composition data as % of fat mass and % of lean mass. Since NT3-injected mice had lower body weight (**Fig 2D**), the % of lean mass was higher based on their lower body weight. When calculated as total fat mass and total lean mass in grams, we found that NT-3 injection reduced total fat mass but did not change total lean mass. To avoid such confusion, we have presented body composition data as fat mass and lean mass in grams in **Fig 2E**.

7. *Was body temperature measured in response to NTF-3 injections?*

We have now measured body temperature in mice with either saline or NT3 injection in response to an acute cold exposure (5°C) and found that NT3-injected mice were more cold-tolerant as they could maintain their body temperature better than saline-injected mice. These data were presented in **Supplemental Figure 4B**, and at page 8, first paragraph.

8. *The proposed NTF3/TRKC signaling-related link between inflammation and neuronal axonal growth should be substantiated by data.*

We have now validated our RNAseq results using real-time RT-PCR approaches showing Cxcl10 expression is significantly up-regulated in sympathetic ganglia from AdNT3-Tg mice, whereas its expression is significantly down-regulated in sympathetic ganglia from TRKC^{+/-} mice after a cold exposure challenge (**Fig 7B-C**). In addition, treating cultured sympathetic neurons treated with recombinant CXCL10 significantly inhibited both basal- and NT-3-stimulated neurite growth (**Fig 7D**), indicating that CXCL10 functionally regulates neuron axonal growth. These data are now described at page 19, third paragraph.

Minor comments:

1. *The authors use both "neurotrophic" and "neurotropic" factor to describe NTF3. Is that on purpose?*

Thanks for pointing this out. "Neurotropic" should be a typo. We have corrected the typo and now all words should read "neurotrophic".

2. *Statements regarding the impact of the findings on obesity treatment should be toned down to a realistic level.*

We agree with the reviewer that the emphasis on the therapeutic significance of our finding might be premature. Thus, we have modified and deleted such statements in the manuscript at page 25, first paragraph.

3. *The sentence: "Although brown fat thermogenesis was traditionally viewed as a defense mechanism against cold in rodents, recent discovery of metabolically active brown fat in adult humans has implicated BAT thermogenesis as a promising therapeutic target for the treatment of obesity" is unclear - why should that be a contradiction?*

Thanks for pointing this out. We have edited this sentence and now it reads: "Brown fat thermogenesis was traditionally viewed as a defense mechanism against cold in rodents. Recent discovery of metabolically active brown fat in adult humans has **further** implicated BAT thermogenesis as a promising therapeutic target for the treatment of obesity". This sentence is located at page 3, first paragraph.

Reviewer 3:

1. *The HFD studies would have been easier to interpret if a lean control had been included. Consequently, we don't really know the level of obesity being induced in these studies.*

Thank you for pointing this out. We have now added body weight data of AdNT3-Tg and WT littermates fed regular chow diet (See **Supplemental Figure 6A**), as well as body weight data of TRKC^{+/-} and WT littermates fed regular diet (See **Supplemental Figure**

18). We found there was no difference in body weight between chow-fed groups measured up to 16 weeks of age, whereas the body weight of AdNT3-Tg and TRKC^{+/-} mice fed HFD became diverging from their littermate controls when measured around 16 weeks of age (~10 weeks on HFD, please see Figures 3I and 8H). Thus, we think our AdNT3-Tg and TRKC^{+/-} mice behave similarly to their WT littermates when fed regular chow diet, but they are resistant to diet-induced obesity when fed HFD. These data were described at page 10, second paragraph, and page 20, second paragraph.

2. *In all the studies the fed glucose level in the blood was presented. It would have been more informative to measure fasted blood glucose. In addition, more repetitive measures during the course of the study would have been more informative. Especially in the AdNT3-Tg mouse study.*

Thanks for the suggestion. We have now measured fasting blood glucose levels in AdNT3-Tg and their WT littermates fed HFD for 6, 8, 12 and 16 weeks (**Supplemental Figure 7A**). We found that fasting glucose level was not different between AdNT3-Tg and WT mice until at a later point on HFD (16 weeks on HFD) (**Supplemental Figure 7A**), when the body weight of AdNT3-Tg mice was significantly lower than that of WT littermates (**Figure 3I**).

In addition, we have also measured glucose and insulin tolerance tests (GTT and ITT, respectively) during the course of HFD feeding in AdNT3-Tg, AdNT3-KI and their respective WT litter mates. We found that whereas GTT and ITT tests were not different between AdNT3-Tg and WT littermates when measured at 6-7 weeks of HFD feeding (**Supplemental Figures 7B-C**) when their body weight was not different, AdNT3-Tg mice did show improved glucose tolerance and insulin sensitivity in GTT and ITT tests measured at 15-16 weeks of HFD feeding (**Supplemental Figures 7D-E**), when AdNT3-Tg mice weighed significantly less than their WT littermates.

Similar results were observed in AdNT3-KI mice. Whereas there was no difference in GTT and ITT between AdNT3-KI and WT littermates when measured at 3-4 weeks on HFD feeding (**Supplemental Figures 12A-B**) when their body weight was not different, AdNT3-KI mice did show improved glucose tolerance and insulin sensitivity in GTT and ITT tests measured 15-16 weeks of HFD feeding (**Supplemental Figures 12C-D**) after they weighed significantly less than their fl/+ littermate controls.

In addition, to study the contribution of liver gluconeogenesis to the regulation of glucose homeostasis, we have also measured pyruvate tolerance test (PTT) in our animal models. We found that there was no difference in PTT tests between AdNT3-Tg and WT littermates measured at either 8 weeks or 14 weeks of HFD feeding (**Supplemental Figures 7F-G**). Further, there was also no difference in PTT test between AdNT3-KI and WT littermates measured at 14 weeks of HFD feeding (**Supplemental Figure 12E**).

Thus, our data indicate that the improved glucose tolerance and insulin sensitivity in mice with adipocyte-specific NT3 overexpression is possibly secondary to their obesity

resistance, and their improved glucose homeostasis is not due to any changes in liver gluconeogenesis, but rather due to improved glucose utilization in the body.

These data are described in the results part at page 11, first paragraph for AdNT3-Tg mice, and page 13, second paragraph.

- 3. The study with the AdNT3-Tg mouse line is probably the most informative aspect of the work. This mouse line used the adiponectin driver for specific expression in adipose tissue. However, how specific is this promoter? The literature reveals a number of studies revealing expression of adiponectin mRNA in skeletal muscle, myocardium and liver. Thus, there is the possibility of over-expression of NT-3 in these tissues. The mouse line should have been fully validated with data on NT-3 expression in these tissues compared between WT and AdNT3-Tg. While systemic expression of NT-3 in serum was not altered there remains the possibility of enhanced NT-3/trkC signaling in these organs/tissues that makes interpretation of the general effects on insulin sensitivity and body weight problematic.*

Thanks for pointing this out. This is an important question that needs to be addressed.

To address this question, we have incorporated the following data into our manuscript:

- 1) We have measured *Nt-3* expression in various tissues of the AdNT3-Tg and WT littermates, including liver, gastrocnemius muscle, and areas of the brain, including arcuate, paraventricular, ventromedial and lateral hypothalamus, cortex and hippocampus. We found no difference in *Nt3* expression in these tissues between AdNT3-Tg and WT littermates, indicating that overexpression of *Nt3* in AdNT3-Tg mice is specific to adipose tissue. Please see **Supplemental Figure 5F**.
- 2) We found no difference in the expression of the neuropeptides pro-opiomelanocortin (*Pomc*), neuropeptide Y (*Npy*), and agouti related neuropeptide (*Agrp*) in arcuate hypothalamus of AdNT3-Tg and WT littermates fed HFD, suggesting that obesity resistance in AdNT3-Tg mice is unlikely mediated through a central mechanism. Please see **Supplemental Figure 6D**.
- 3) To assess whether liver gluconeogenesis could contribute to improved glucose homeostasis in AdNT3-Tg mice, we performed pyruvate tolerance test (PTT) in HFD-fed AdNT3-Tg and WT mice. However, we did not observe any differences in PTT between AdNT3-Tg and WT mice when measured at either 8 weeks or 14 weeks of HFD feeding, indicating that improved glucose homeostasis in AdNT3-Tg mice is not due to changes in liver gluconeogenesis, but rather due to improved glucose utilization in the body. Please see **Supplemental Figures 7F-G**.

These data are described in the results part at page 9, first paragraph, page 10, second paragraph and page 11, first paragraph.

- 4. The authors tried to complement the AdNT3-Tg study with an alternative mouse line, the AdNT3-KI mouse. This is laudable, however, the issue here is the very high systemic*

levels of NT-3 in these mice (Supp Fig. 7 E). This finding makes interpretation of the insulin sensitivity and body weight data very difficult (e.g. ability to link the findings to changes in cold-induced thermogenesis).

Thank for pointing this out. We have addressed these questions by incorporating the following data into the manuscript:

- 1) We have measured *Nt-3* expression in various tissues of the AdNT3-KI and fl/+ littermates, including liver, gastrocnemius muscle, and areas of the brain, including arcuate, paraventricular, ventromedial and lateral hypothalamus, cortex and hippocampus. We found no difference in *Nt3* expression in these tissues between AdNT3-KI and fl/+ littermates, indicating that overexpression of *Nt-3* in AdNT3-KI mice is specific to adipose tissue. Please see **Supplemental Figure 8E**.
- 2) We found no difference in the expression of the neuropeptides pro-opiomelanocortin (*Pomc*), neuropeptide Y (*Npy*), and agouti related neuropeptide (*AgRP*) in arcuate hypothalamus of AdNT3-KI and fl/+ littermates fed HFD, suggesting that obesity resistance in AdNT3-KI mice is unlikely mediated through a central mechanism. Please see **Supplemental Figure 11A**.
- 3) To assess the contribution of skeletal muscle and liver to the obesity resistance phenotypes in AdNT3-KI mice, we have also measured gene expressions in gastrocnemius muscle and liver. However, there was also no difference in the expression of genes associated with mitochondria function and lipid metabolism in muscle and liver between AdNT3-KI and fl/+ mice, indicating that the obesity resistance observed in AdNT3-KI mice may be primarily due to increased thermogenesis in adipose tissue, but not in muscle and liver. Please see **Supplemental Figures 11B-C**.
- 4) To assess whether liver gluconeogenesis could contribute to improved glucose homeostasis in AdNT3-KI mice, we performed pyruvate tolerance test (PTT) in HFD-fed AdNT3-KI and fl/+ mice. However, we did not observe any differences in PTT between AdNT3-KI and fl/+ mice when measured at 14 weeks of HFD feeding, indicating that improved glucose homeostasis in AdNT3-KI mice is not due to changes in liver gluconeogenesis, but rather due to improved glucose utilization in the body. Please see **Supplemental Figures 12E**.

These data are described in the results part at page 12, first paragraph, page 12, last paragraph to page 13, first and second paragraph.

5. *Interpretation of findings from the *trkC* +/- or STRKCKO mice reveals the same drawbacks as above. Here systemic effects on *trkC*/NT-3 signaling could contribute to energy metabolism and insulin sensitivity at multiple sites and thus the link with proposed effects of *trkC* signaling and adipose tissue and cold-induced thermogenesis is indistinct.*

Thanks for pointing this out. We have previously measured norepinephrine (NE) content and NE turnover rate (NETO) in brown and white adipose tissues of TRKC^{+/-} and WT littermate controls after a 16-hour cold challenge. We found that NE content and NETO

were significantly higher in brown and white adipose tissue of TRKC^{+/-} than that of WT mice (**Supplemental Figures 16A-B**).

To address the reviewer's comments, we have extended our findings and measured NE content and NETO in other tissues (skeletal muscle, kidney and heart) in TRKC^{+/-} and WT mice after the 16-hour cold challenge. Please see **Supplemental Figures 16C-D**. Interestingly, we found there was no difference in NE and NETO between TRKC^{+/-} and WT mice after the 16-hour cold exposure. These data indicate that the reduced sympathetic activity in TRKC^{+/-} mice is probably specific to adipose tissues, without affecting sympathetic activities in other tissues.

This is also partially supported by our findings presented in **Figure 6A-C**, where we studied the co-localization of sympathetic marker tyrosine hydroxylase (TH) and TRKC in neurons innervate iBAT (fast blue FB+) or neurons innervate other tissues (FB-) during cold exposure. Interestingly, we found cold exposure specifically increased sympathetic neurons (TH+TRKC+) that innervate iBAT (TH+TRKC+FB+), but did not affect the neurons innervating other tissues (TH+TRKC+FB-).

Thus, these data indicate that TRKC-regulated signaling in sympathetic ganglia neurons is probably specific to adipose tissue. These data are described in the results part at page 17, third paragraph.

6. *Which sympathetic ganglia were cultured in Figure 4 and were they adult neurons?*

The neurons used for primary culture and neurite growth experiments in **Figure 4 and 5** in the revised manuscript were isolated from thoracic T1-T4 levels (that innervate iBAT) from 6-8-week-old adult mice. We have added this information under the figure legends of Figures 4 and 5, and in the Methods part at page 33, second paragraph.

7. *In Fig.4 for the cultures of WT vs trkC +/- were the neurons exposed to NT-3?*

Since neurons from TRKC^{+/-} mice would have one allele of TRKC expression, to study whether TRKC is necessary for NT3's neurotrophic effects, we isolated sympathetic neurons from fl/fl and STRKCKO mice, where TRKC is specifically deleted in TH-positive sympathetic neurons, and treated these neurons with either phosphate-buffered saline (PBS) or NT-3. We found that deleting TRKC in sympathetic neurons from STRKCKO mice significantly inhibited basal- and NT-3-stimulated neurite growth, indicating TRKC is required for NT-3's neurotrophic effects. Please see **Figure 5B** in the revised manuscript. The data are described in the results part, page 15, second paragraph.

8. *In Figure 5 it is not clear how the numbers of trkC positive neurons innervating iBAT from T2 ganglia is changing. Not discussed anywhere. Is the suggestion that trkC positive*

*neurons not normally innervating the iBAT now innervate upon cold exposure? Or are normally *trkC* negative neurons becoming *trkC* positive in response to cold?*

Thanks for pointing this out. In the revised manuscript, it is now **Figure 6B-C**. We did observe the increase in TH⁺TRKC⁺FB⁺ triple-labeled neurons in T2 sympathetic ganglia with concomitant decrease in TH⁺ single labeled and TH⁺TRKC⁺ double labeled neurons. We think that during cold exposure, TH⁺ single labeled and TH⁺TRKC⁺ double labeled neurons might gain the expression of TRKC, and since NT3/TRKC signaling stimulates neurite growth, increased TRKC expression in these neurons may have stimulated axonal growth that resulted in increased innervation specifically to fat tissues. We have incorporated such discussion in the results part, page 16, first paragraph.

9. *In Fig.5 (E) the % of total TH +ve or cFos +ve neurons innervating the iBAT are much higher than in the study depicted in parts A-D. This seems odd since they are actually *trkC* +/-.*

Thanks for pointing this out. We have added some clarification in the Figure legends. These figures are now in **Figure 6**. In **Figure 6B**, the % of TH⁺, TH⁺FB⁺, and TH⁺TRKC⁺FB⁺ neurons are calculated as mutually exclusive numbers, i.e., TH⁺ single-labeled neuron calculation does not include the number of TH⁺FB⁺ double labeled and TH⁺FB⁺TRKC⁺ triple labeled neurons. Likewise, the TH⁺FB⁺ double labeled neuron calculation does not include the number of TH⁺FB⁺TRKC⁺ triple labeled neurons. Thus, the TH⁺FB⁺ neurons in **Figure 6E and 6G** are actually the sum of TH⁺FB⁺ and TH⁺FB⁺TRKC⁺ neurons in **Figure 6B**, which should be roughly similar. We have added this clarification in the description of Figure legends under **Figure 6B**.

Reviewers' Comments:

Reviewer #1:

Remarks to the Author:

The authors discover a novel fat-derived "adipokine" NT-3 in regulating SNS growth and innervation in adipose tissue, which induces energy expenditure and improves glucose metabolism. They used several mice models to confirm their discovery, such as NT-3 transgenic mice, TRKC^{+/-} mice and SNS specific TRKC KO mice. This study is of great interest in its field and provides a new target for type 2 diabetes and obesity.

The revised manuscripts accepted my suggestions and answered my questions very well. However, I still have some minor comments:

1. Formally, genes should be italic and with initial letter capitalized.
2. There are no K and L panels in Fig.S10, but they are described in figure legends.
3. If possible, the author can add a working model that simply summarize the current work.

Reviewer #2:

Remarks to the Author:

The authors responded very well to almost all of my critical points and questions. The manuscript adds novel and timely data to an emerging research field. The response to the question how beige and brown adipocytes were distinguished did not convince me entirely, but that should not limit my overall support for the revised manuscript.

Reviewer #3:

Remarks to the Author:

My concerns with this manuscript have been addressed in a comprehensive manner. New data has been added that significantly strengthens the conclusions made with regard to the transgenic mouse models utilized.

Reviewer #1 (Remarks to the Author):

The authors discover a novel fat-derived “adipokine” NT-3 in regulating SNS growth and innervation in adipose tissue, which induces energy expenditure and improves glucose metabolism. They used several mice models to confirm their discovery, such as NT-3 transgenic mice, TRKC^{+/-} mice and SNS specific TRKC KO mice. This study is of great interest in its field and provides a new target for type 2 diabetes and obesity.

The revised manuscripts accepted my suggestions and answered my questions very well. However, I still have some minor comments:

1、 Formally, genes should be italic and with initial letter capitalized.

Response: Thank you for pointing this out. We have carefully checked and made sure gene names are italic with initial letter capitalized in both the main text and figures.

2、 There are no K and L panels in Fig.S10, but they are described in figure legends.

Response: Thank you for pointing this out. This is a typo. We have removed the description of K and L in Fig S10.

3、 If possible, the author can add a working model that simply summarize the current work.

Response: A schematic illustration is added in Fig 10.

Reviewer #2 (Remarks to the Author):

The authors responded very well to almost all of my critical points and questions. The manuscript adds novel and timely data to an emerging research field. The response to the question how beige and brown adipocytes were distinguished did not convince me entirely, but that should not limit my overall support for the revised manuscript.

Response: Thank you for your constructive comments.

Reviewer #3 (Remarks to the Author):

My concerns with this manuscript have been addressed in a comprehensive manner. New data has been added that significantly strengthens the conclusions made with regard to the transgenic mouse models utilized.

Response: Thank you for your constructive comments.